# Integrating hydrogen utilization in $CO_2$ electrolysis with reduced energy loss

Xiaoyi Jiang [1,2], Le Ke [1,2], Kai Zhao [1,2], Xiaoyu Yan [1], Hongbo Wang [1], Xiaojuan Cao [1], Yuchen Liu [1,2], Lingjiao Li [1,2], Yifei Sun [3], Zhiping Wang [1], Dai Dang [4] & Ning Yan [1,2] ✉

Electrochemical carbon dioxide reduction reaction using sustainable energy is a promising approach of synthesizing chemicals and fuels, yet is highly energy intensive. The oxygen evolution reaction is particularly problematic, which is kinetically sluggish and causes anodic carbon loss. In this context, we couple $CO_2$ electrolysis with hydrogen oxidation reaction in a single electrochemical cell. A $Ni(OH)_2$/NiOOH mediator is used to fully suppress the anodic carbon loss and hydrogen oxidation catalyst poisoning by migrated reaction products. This cell is highly flexible in producing either gaseous (CO) or soluble (formate) products with high selectivity (up to 95.3%) and stability (>100 h) at voltages below 0.9 V (50 mA cm$^{-2}$). Importantly, thanks to the "transferred" oxygen evolution reaction to a water electrolyzer with thermodynamically and kinetically favored reaction conditions, the total polarization loss and energy consumption of our $H_2$-integrated $CO_2$ reduction reaction, including those for hydrogen generation, are reduced up to 22% and 42%, respectively. This work demonstrates the opportunity of combining $CO_2$ electrolysis with the hydrogen economy, paving the way to the possible integration of various emerging energy conversion and storage approaches for improved energy/cost effectiveness.

Electrocatalytic $CO_2$ reduction reaction ($CO_2$RR) at low temperatures is among the most promising approaches of sustainably producing fuels and chemicals[1–3]. Tremendous progress regarding materials development, fundamental understandings and technological innovations have been made over the past decades, rendering high-rate and selective formation of products spanning from C1 (e.g., CO and CH$_4$) to C3 (e.g., propanol)[4–12]. However, the high energy input and low energy efficiency remain a major barrier impeding the large-scale application in real life. Specifically, the oxygen evolution reaction (OER) at the anode is both energetically intensive and kinetically sluggish, yet generating diatomic oxygen with low market value[13–17]. The oxygen production often causes carbon loss, further reducing the energy efficiency[18–20]: the readily formed carbonate or bicarbonate ions at the cathode can migrate, through either aqueous electrolyte or anion-exchange membrane (AEM), to the anode with a lower local pH value. Consequently, the protonation results in the release of $CO_2$ together with the evolved $O_2$. In a typical AEM based $CO_2$ electrolyzer, carbon loss caused by $CO_2$ crossover could reach ca.70%[21]. The energy penalty of recovering $CO_2$ using amine scrubbing is gigantic, ca. 3-7 GJ/tonne $CO_2$[19,20]. Recent estimations show that this energy is even ~1.6 times more demanding than the electrolysis step[20,21].

In this context, the so-called "paired electrolysis" becomes a hot research topic, in which $CO_2$RR is coupled with a thermodynamically and/or kinetically more favorable half-reaction[16,22–25]. Electro-oxidations of aldehyde, glycerol, ethanol, isopropanol, 1,2-propanediol and other organics have been identified as suitable OER

[1]School of Physics and Technology, Wuhan University, Wuhan 430072, China. [2]Shenzhen Research Institute of Wuhan University, Shenzhen 518057, China. [3]Shenzhen Research Institute of Xiamen University, Shenzhen 518057, China. [4]School of Chemical Engineering and Light Industry, Guangdong University of Technology, Guangzhou 510006, China. ✉e-mail: ning.yan@whu.edu.cn

substitutes, which indeed lower both energy input and overpotential loss while coproducing value-added products[16,25–28]. Nonetheless, this strategy has two potential limitations: the prominent one is the huge market-size mismatch between the cathode and anode chemicals. For instance, the emitted $CO_2$ from the industry sector alone that can be captured and utilized is in gigatonne level per year, yet the annual market demand of nearly all products which can be potentially coproduced from the paired electrolysis is in (and often less than) the million-tonne scale[29–31]. Moreover, the separation and purification of products from the electrolyte pose another challenge toward the practical application[24].

On the big picture of renewable energy storage, hydrogen is another major energy carrier for various downstream applications[32]. Pairing $CO_2RR$ with hydrogen oxidation can theoretically address the abovementioned challenges while reducing the energy input. It should be noted that green hydrogen generation via water electrolysis also involves the OER which can occur at much more favored thermodynamic and kinetic conditions compared with that in $CO_2RR$ reactors (e.g., neutral or weaker-alkaline electrolytes and lower operating temperatures)[13,33–36]. For instance, the anodic overpotential loss in advanced solid oxide water electrolysis cell (SOEC) and alkaline water electrolyzer (AWE) is as low as 0.01 V and 0.2 V at 50 mA cm$^{-2}$, respectively; yet, it often exceeds 0.52 V in $CO_2RR$ reactor at neutral conditions[35,37]. One would imagine intuitively if we can "transfer" the OER in $CO_2RR$ to water electrolyzer to boost the energy efficiency.

Inspired by the facts above, we proposed the direct coupling of $CO_2$ electrolysis with hydrogen oxidation reaction (HOR) at the anode in a single electrochemical cell. Using $CO_2$-to-CO and $CO_2$-to-formate as two model reactions to respectively represent the reactions yielding gaseous and electrolyte-soluble products, we showed the effectiveness and efficiency of such $H_2$-integrated $CO_2RR$. A flow cell with a Ni(OH)$_2$/NiOOH mediator was designed to prevent carbon loss and HOR catalyst poisoning by the migrated $CO_2RR$ products; we also developed a gas-diffusion electrode with a gradient functional layer which minimized the cathodic overpotential loss. The kinetic advantages were discussed in comparison with conventional systems. Importantly, our $H_2$-integrated $CO_2RR$, coupled with either AWE or SOEC, promised to decrease the total energy consumption up to 42%.

## Results

### Theoretical considerations and cell design

Figure 1a shows the conventional $CO_2RR$ process in comparison with the $H_2$-integrated $CO_2RR$ combining with water electrolysis. The problematic OER is transferred to a water electrolyzer, potentially alleviating the high energy consumption in conventional $CO_2RR$ brought by carbon loss and high OER overpotential. Note that pairing $CO_2$ with HOR is also practically feasible in terms of "mass balance": assuming 100 million tonnes of stored $H_2$ (approximately the global $H_2$ production in 2021) is used to convert $CO_2$ to CO, 2.2 gigatonnes of $CO_2$ can be stoichiometrically utilized, accounting for ~25% of global $CO_2$ emissions from the industry sector in 2022 according to the

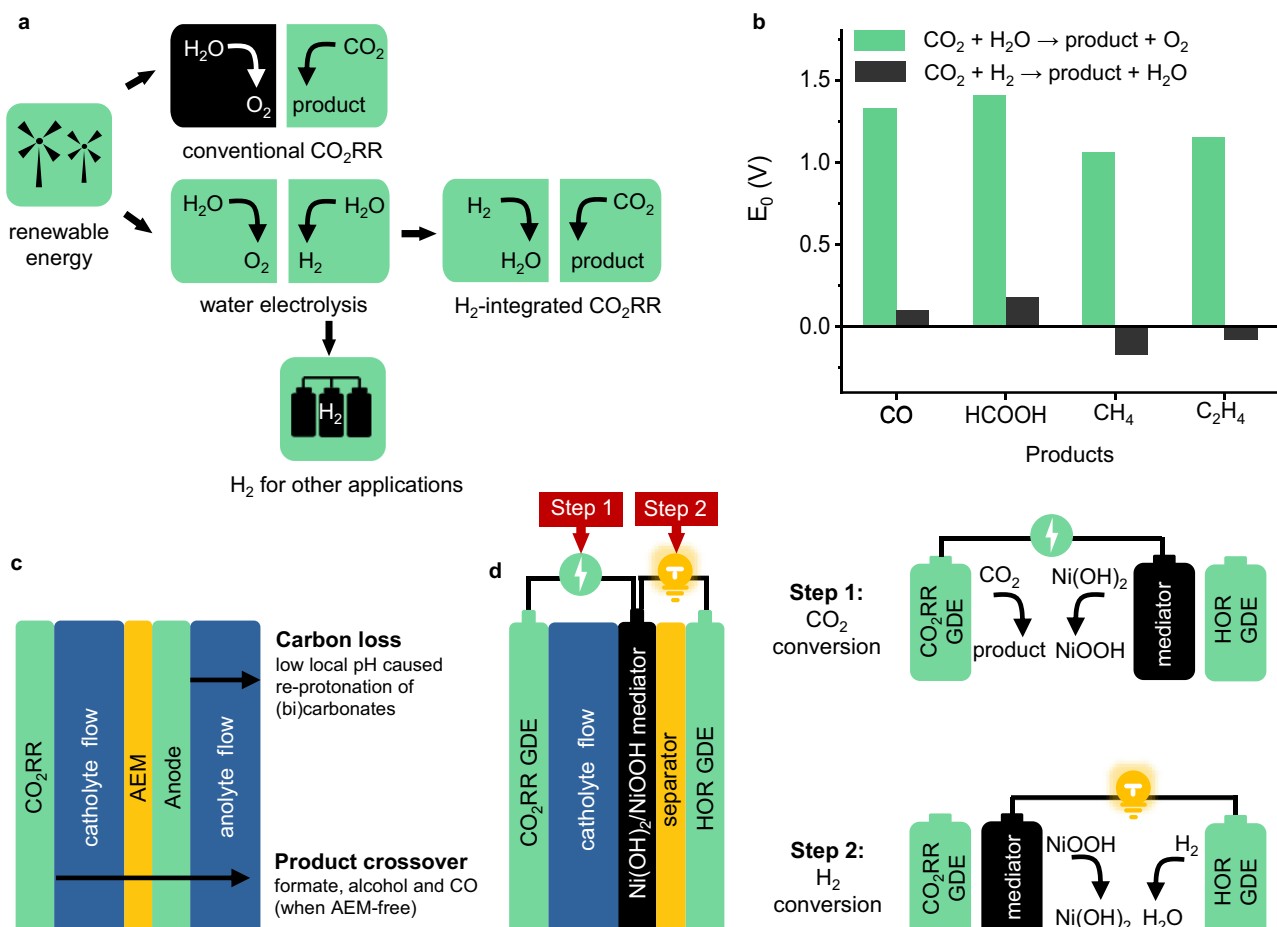

**Fig. 1 | $H_2$-integrated $CO_2RR$ and the cell configuration. a** Illustrative comparison of $H_2$-integrated and conventional $CO_2RR$; **b** Comparison of Nernst potentials ($E_0$) between conventional and $H_2$-integrated $CO_2RR$; **c** Potential carbon loss and product crossover at the anode side in a typical $CO_2$ flow electrolyzer; **d** Cell configuration and detailed working principles of the $H_2$-integrated $CO_2RR$ cell.

International Energy Agency (IEA). Figure 1b compares the Nernst potentials of conventional and $H_2$-integrated $CO_2RR$ at standard conditions in four typical conversions. Reactions toward CO and formic acid remain electrolytic in $H_2$-integrated $CO_2RR$, yet the Nernst potentials drop to only 0.10 and 0.18 V, respectively. Reactions toward $CH_4$ and $C_2H_4$ are even more promising, as the potentials respectively decrease to −0.17 and −0.08 V, implying that these conversions can theoretically generate electricity (see Supplementary Table 1 in the Supplementary Information for details). One would envisage that using a classic flow reactor or a membrane electrode assembly (MEA) would realize this concept. But there are at least two catches as shown in Fig. 1c: (1) the low local pH at the anode will lead to the reprotonation of migrated (bi)carbonate ions and the release $CO_2$ gas. It mixes with the residual $H_2$ in the anode effluent, causing significant $CO_2$ loss; (2) $CO_2RR$ products such as CO (when membrane-free), formate and alcohols can migrate through the (membrane) electrolyte, potentially deactivating HOR catalysts (*vide infra*).

To achieve effective and efficient $H_2$-integrated $CO_2RR$, we designed a single cell with the detailed configuration shown in Fig. 1d and examined its effectiveness by performing $H_2$-integrated $CO_2$-to-CO and $CO_2$-to-formate as two model reactions. The cell contains a $Ni(OH)_2/NiOOH$ mediator, sandwiched by a $CO_2RR$ gas diffusion electrode (GDE) and a HOR GDE, to decouple $CO_2RR$ and HOR while eliminating the sluggish oxygen catalysis. The working principle of the cell contains temporally and spatially separated Step 1 and Step 2, featuring an alternating operating sequence. Step 1 is for $CO_2$ conversion in which $CO_2$ is reduced to CO or formate at $CO_2RR$ GDE and $Ni(OH)_2$ is oxidized to NiOOH as shown in Eqs. (1)–(5):

cathode:

$$CO_2 + H_2O + 2e^- \rightarrow CO + 2OH^- \tag{1}$$

$$CO_2 + 2H_2O + 2e^- \rightarrow HCOOH + 2OH^- \tag{2}$$

anode:

$$Ni(OH)_2 + OH^- \rightarrow NiOOH + H_2O + e^- \tag{3}$$

overall:

$$CO_2 + 2Ni(OH)_2 \rightarrow 2NiOOH + CO + H_2O \tag{4}$$

$$CO_2 + 2Ni(OH)_2 \rightarrow 2NiOOH + HCOOH \tag{5}$$

Compared with OER half reaction, the oxidation of $Ni(OH)_2$ is a single-electron transfer process with much lower overpotential loss (*vide infra*).

After the consumption of $Ni(OH)_2$, Ni electrode then works together with HOR GDE in Step 2 for energy harvesting with reactions shown below in Eqs. (6)–(8):

cathode:

$$NiOOH + H_2O + e^- \rightarrow Ni(OH)_2 + OH^- \tag{6}$$

anode:

$$H_2 + 2OH^- \rightarrow 2H_2O + 2e^- \tag{7}$$

overall:

$$2NiOOH + H_2 \rightarrow 2Ni(OH)_2 \tag{8}$$

This step is the typical reaction in a Ni-$H_2$ battery, generating electricity to partially compensate the power consumed in Step 1. The continuous operation of the system is enabled by the periodical swap between Step 1 and Step 2.

## Reaction kinetics of electrodes

In the $CO_2$-to-CO model reaction, we used Zn nanosheet, prepared by electrodeposition, as the catalyst[38–40]. The electrodeposition time is regulated (t = 0 s, 100 s, 500 s, and 1000 s) and the resulting samples are donated as Cu, Zn-Cu-100, Zn-Cu-500, and Zn-Cu-1000, respectively. The X-ray diffraction (XRD) pattern of Cu foam (Fig. 2a) shows characteristic diffraction peaks of pure metallic Cu (JCPDS No.04-0836). The extra peaks at 42.8° and 48.7° are attributed to $Cu_2O$ (JCPDS No.05-0667) and CuO (JCPDS No.48-1548), respectively, as Cu is readily oxidized in ambient air. After Zn electrodeposition, two new peaks located at 36.3° and 38.9° appear, which are assigned to metallic Zn (JCPDS No.87-0713). Their intensity increases with the electrodeposition time because of the increased Zn loading (see Supplementary Table 2). The scanning electron microscopy (SEM) images show that all the deposited Zn has a nanosheet structure (see Fig. 2c and Supplementary Fig. 1b–d). The thickness of these nanosheets increases from 3 to 5 nm for Zn-Cu-100 to *ca.*50 nm for Zn-Cu-1000. The length of each sheet also grows from ∼100 nm for Zn-Cu-100 to ∼1.5 μm for Zn-Cu-1000.

The $CO_2RR$ activity of as-synthesized electrocatalysts was first evaluated in a classic H-cell using $CO_2$-saturated 0.1 M $KHCO_3$ as the electrolyte. The linear sweep voltammetry (LSV) of Cu foam electrode shows a low current density, implying low $CO_2RR$ activity (see Supplementary Fig. 2). Zn electrodeposition leads to increased current densities. The highest current density is achieved on Zn-Cu-500 electrode, which is mainly attributed to the largest surface area (Zn-Cu-100: 9.9208 $m^2$ $g^{-1}$; Zn-Cu-500: 25.6896 $m^2$ $g^{-1}$; Zn-Cu-1000: 24.3713 $m^2$ $g^{-1}$, see nitrogen adsorption-desorption isotherms in Supplementary Fig. 3). The smaller surface area of Zn-Cu-1000 could be ascribed to the "overgrowth" of Zn nanosheets as evidenced by SEM images. Pore size distribution analysis in Supplementary Fig. 4 confirms that Zn-Cu-500 also has the largest pore volume of 0.035 $cm^3$ $g^{-1}$. The electrochemical surface areas (ECSAs) are consistent with these results, as the largest ECSA comes from Zn-Cu-500 (see Supplementary Figs. 5 and 6).

$CO_2RR$ products from these catalysts were monitored during potentiostatic electrolysis in the potential window ranging from −0.5 to −1.1 V (vs. RHE and hereafter). Zn catalysts show completely different behaviors in comparison with Cu foam (see Fig. 2d and Supplementary Figs. 7 and 8 for details). In particular, Zn-Cu-500 exhibits the highest Faradaic efficiency (FE) toward CO formation (85.8% at -1.0 V, also see Supplementary Figs. 7 and 8), while the partial current density for CO ($j_{CO}$) reaches *ca.* 11 mA $cm^{-2}$ (see Fig. 2e). This performance is comparable with the state-of-the-art CO-selective catalysts synthesized using sophisticated approaches[38,40]. Further decreasing the applied potential, however, leads to a drop of CO FE, which is attributed to mass-transport limitations at high current densities. The activity difference among three catalysts pertains to the variance of number of active sites rather than the change of intrinsic activity, since the surface-area-normalized current densities at low overpotentials for all catalysts are similar (see Supplementary Fig. 9). Note that the normalized current density for Zn-Cu-500 at high overpotentials was lower than that of Zn-Cu-100. This suggests that the complex nanostructure, while offering higher surface area and more active sites, might have mass transfer problems.

In the $CO_2$-to-formate model reaction, we also utilized a nanostructured electrocatalyst based on $Bi_2O_3$. In the hydrothermal synthesis, porous carbon nanospheres are added as the template on which Bi species are deposited. A final calcination step burns away carbon templates and converts Bi species into $Bi_2O_3$ porous nanospheres (see characterization results in Supplementary Fig. 10). This catalyst is

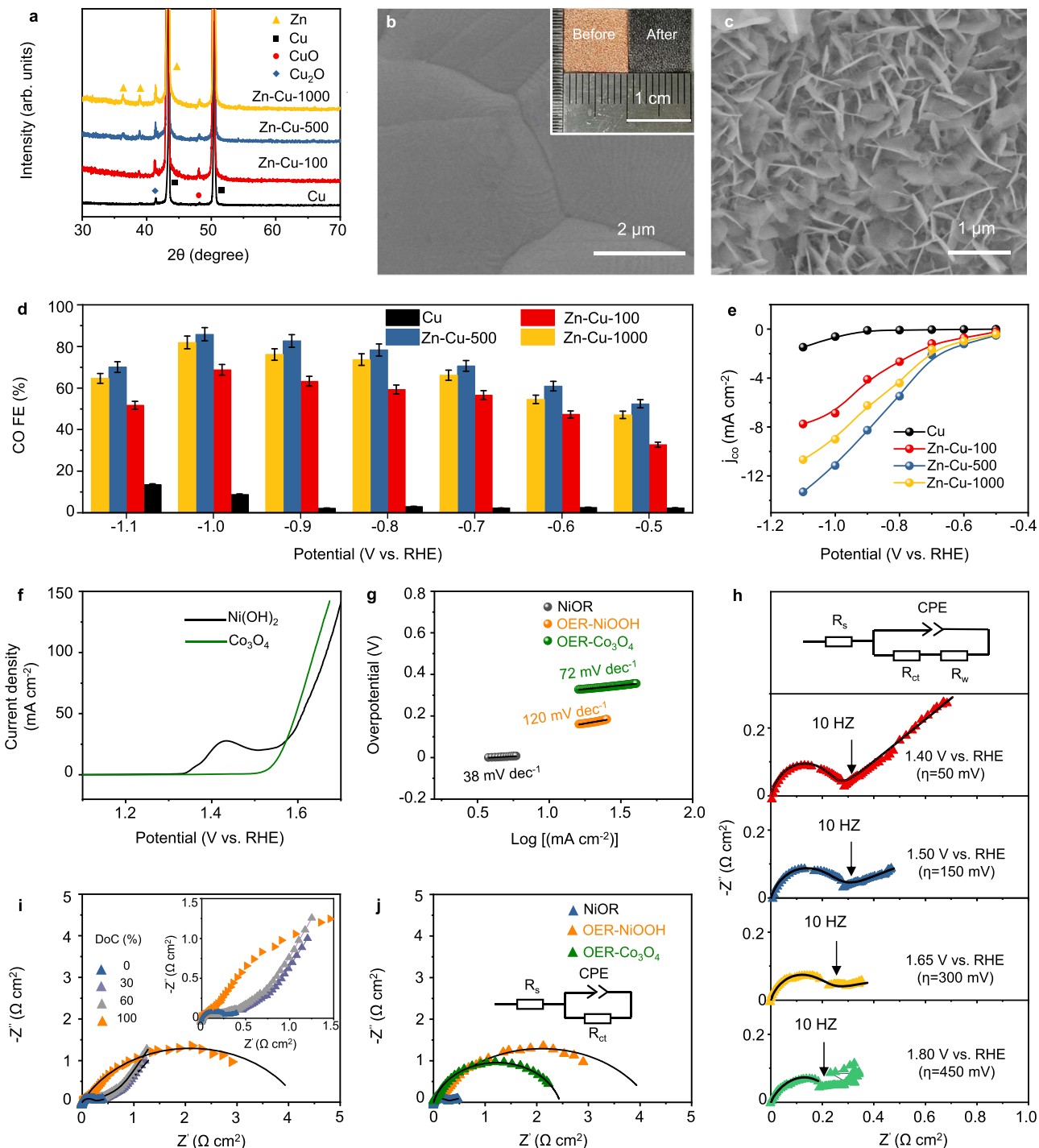

**Fig. 2 | Characterizations and performance of Zn and Ni(OH)₂/NiOOH electrodes. a** XRD patterns; SEM images of **b** Cu foam and **c** Zn-Cu-500, the inset compares the optical images of Cu foam before and after Zn deposition; **d** Potential-dependent Faradaic efficiency for CO generation (the error bar represents standard deviation from three independent measurements); **e** Plots of $j_{CO}$ as a function of potential bias during $CO_2RR$; **f** LSV curves of $Ni(OH)_2$ and $Co_3O_4$ electrodes (the solution resistance was 1.25 - 1.35 Ω); **g** Tafel plots of NiOR and OER; **h** Nyquist plots of $Ni(OH)_2$ electrode acquired at 1.40, 1.50, 1.65, and 1.80 V vs. RHE; **i** Nyquist plots of $Ni(OH)_2/NiOOH$ electrode acquired at different DoC at 1.55 V vs. RHE, charging is defined as the $Ni(OH)_2$-to-NiOOH conversion; **j** Nyquist plots measured at 1.55 V vs. RHE of $Ni(OH)_2/NiOOH$ electrode (DoC = 0% and =100%) and $Co_3O_4$. All solid black lines in Nyquist plots represent the fitting results based on the equivalent circuit, where $R_s$ is the solution resistance, CPE is the constant phase element, $R_{ct}$ is the charge-transfer resistance and $R_W$ is the Warburg impedance. Source data are provided as a Source Data file.

highly selective to yield formate in the potential range from −0.45 to −1.05 V. The top formate FE reaches 89.0% at −0.65 V as shown in Supplementary Figs. 11 and 12.

We also examined the reaction kinetics of $Ni(OH)_2/NiOOH$ mediator. Although its electrochemical behaviors in strong alkaline media are well documented in the literature[41–44], the performance in weaker base, particularly in comparison with OER, is rarely reported. In the cyclic voltammogram (CV) shown in Supplementary Fig. 13, the redox peaks, corresponding to the $Ni^{2+}/Ni^{3+}$ conversion, are clearly observable. The onset potential of OER is -220 mV higher than that of

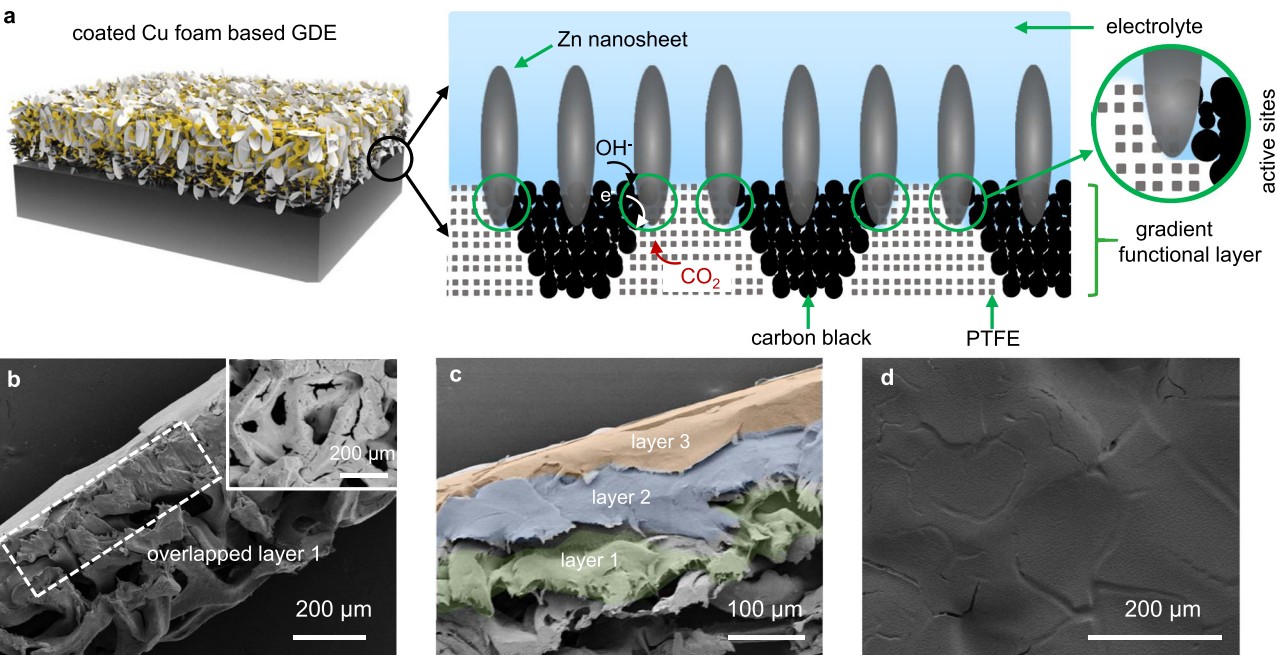

**Fig. 3 | Characterizations of GDE based on the Zn-coated Cu foam. a** The schematic illustration of GDE; cross-sectional SEM images of **b** merged foam with carbon-PTFE layer after drop-casting the first layer (inset shows the surface image) and **c** final gradient functional layer; **d** SEM image of the top layer surface.

Ni(OH)$_2$ oxidation reaction (NiOR). As the Nernst potential of NiOR is higher than that of OER, the lower onset potential suggests the much-favored reaction kinetics as OER is a concerted four-electron-proton transfer process. This conclusion is also valid when comparing with a state-of-the-art OER catalyst, namely, nanostructured Co$_3$O$_4$ (detailed materials characterizations and OER performance are shown in Supplementary Fig. 14)[45], since NiOR shows lower onset potential and Tafel slope (see Fig. 2f, g).

The dynamic behaviors of Ni(OH)$_2$ electrode during NiOR are further examined using electrochemical impedance spectroscopy (EIS). The Nyquist plots, together with the fitting results, in Fig. 2h reveal charge-transfer kinetics at various potentials. Both the charge-transfer resistance (R$_{ct}$) and the Warburg impedance (R$_w$) decrease progressively when the potential bias increases from the open-circuit potential (OCP) to 1.65 V vs. RHE, implying that the NiOR kinetics is controlled by both charge transfer and mass transfer. Note that further increasing the potential bias causes the rise of R$_{ct}$ due to the initiation of OER (also see Supplementary Table 3). The composition of Ni(OH)$_2$ electrode also affects reaction kinetics. In the Nyquist plots where the depth-of-charge (DoC) of Ni(OH)$_2$ varies (see Fig. 2i and Supplementary Table 4, charging is defined as Ni(OH)$_2$-to-NiOOH conversion), R$_{ct}$ remains essentially identical when DoC is 0%, 30% and 60%. However, it drastically increases from 0.20 Ω to 4.19 Ω cm$^2$ when DoC reaches 100%, implying that the charge transfer process was then dominated by OER. Compared with the R$_{ct}$ of OER on either NiOOH or Co$_3$O$_4$, the R$_{ct}$ of NiOR is more than an order of magnitude smaller as shown in Fig. 2j and Supplementary Table 5. We also characterized the reverse process, i.e., NiOOH reduction reaction (NiRR), as shown in Supplementary Fig. 15. It seems that the diffusion process becomes the rate-limiting step: when the overpotential increases, R$_{ct}$ remains nearly constant while R$_w$ shows substantial rise (see Supplementary Table 6).

In the steady-state study using chronopotentiometric method (±10 mA cm$^{-2}$), stable redox reactions are observed in Supplementary Fig. 16. The high reversibility of Ni electrode in weak KOH electrolyte is also reflected by the roughly identical charge (*ca.* 200 C) stored/released in the reactions. The calculated Coulombic efficiency is >99%. Before the complete consumption of Ni(OH)$_2$ in the oxidation cycle, oxygen bubbles are hardly observable, implying that a simultaneous

and active OER is unlikely (*cf.* EIS spectra with different DoC and additional experimental proof below). Bubbles starts to appear when all Ni(OH)$_2$ is converted into NiOOH (see the comparison in the inset of Supplementary Fig. 16), accompanied by a sudden potential increase from 1.45 to 1.54 V. Hence we confirm that Ni(OH)$_2$/NiOOH mediator could perform well in weak alkaline electrolyte.

## Cell performance

The detailed cell assembly for H$_2$-integrated CO$_2$RR is shown in Supplementary Fig. 17. Bi$_2$O$_3$ catalyst can be easily incorporated into the conventional carbon based GDE via infiltration. However, transforming Zn-Cu-500 foam into a GDE is not such straightforward. We thus developed a gradient functional layer, as illustrated in Fig. 3a, which was fabricated via a "layer-by-layer" method to form the GDE where the maximized amounts of active sites, effective mass transport and adequate mechanical strength with good conductivity are guaranteed concurrently. The gradient microstructure contains gas diffusion channels in the polytetrafluoroethylene (PTFE) phase and ion/electron transport pathways (carbon phase), which enables effective "inward" transport of CO$_2$ and "outward" transport of electrolyte to the active sites. In addition, it simultaneously allows lower contact resistance with the Zn-Cu-500 foam (due to higher content of carbon in the inner layer) and sufficient mechanical strength (due to higher content of PTFE in the outer layer). The gradient functional layer partially overlaps with the porous foam because of the slurry-drop-casting fabrication technique. The electrochemically active sites, where Zn nanosheets share common areas with carbon, PTFE and aqueous electrolyte, are thus created to include more Zn nanosheets.

The SEM images confirm the successful preparation of the gradient functional layer in the GDE. Figure 3b demonstrates the cross-sectional image of Zn-Cu-500 foam after applying the first layer. It seems that all the drop-casted carbon-PTFE composite stays inside the porous structure without forming an intact surface film (see the inset in Fig. 3b). After applying another two layers with varied carbon-PTFE ratios, a robust, dense and water-proof film finally appears on the top. Neither delamination among layers nor cracks/pores on the surface is observed in the SEM images in Fig. 3c, d and Supplementary Fig. 18. Moreover, XRD patterns of GDE indicate the phase stability of Zn

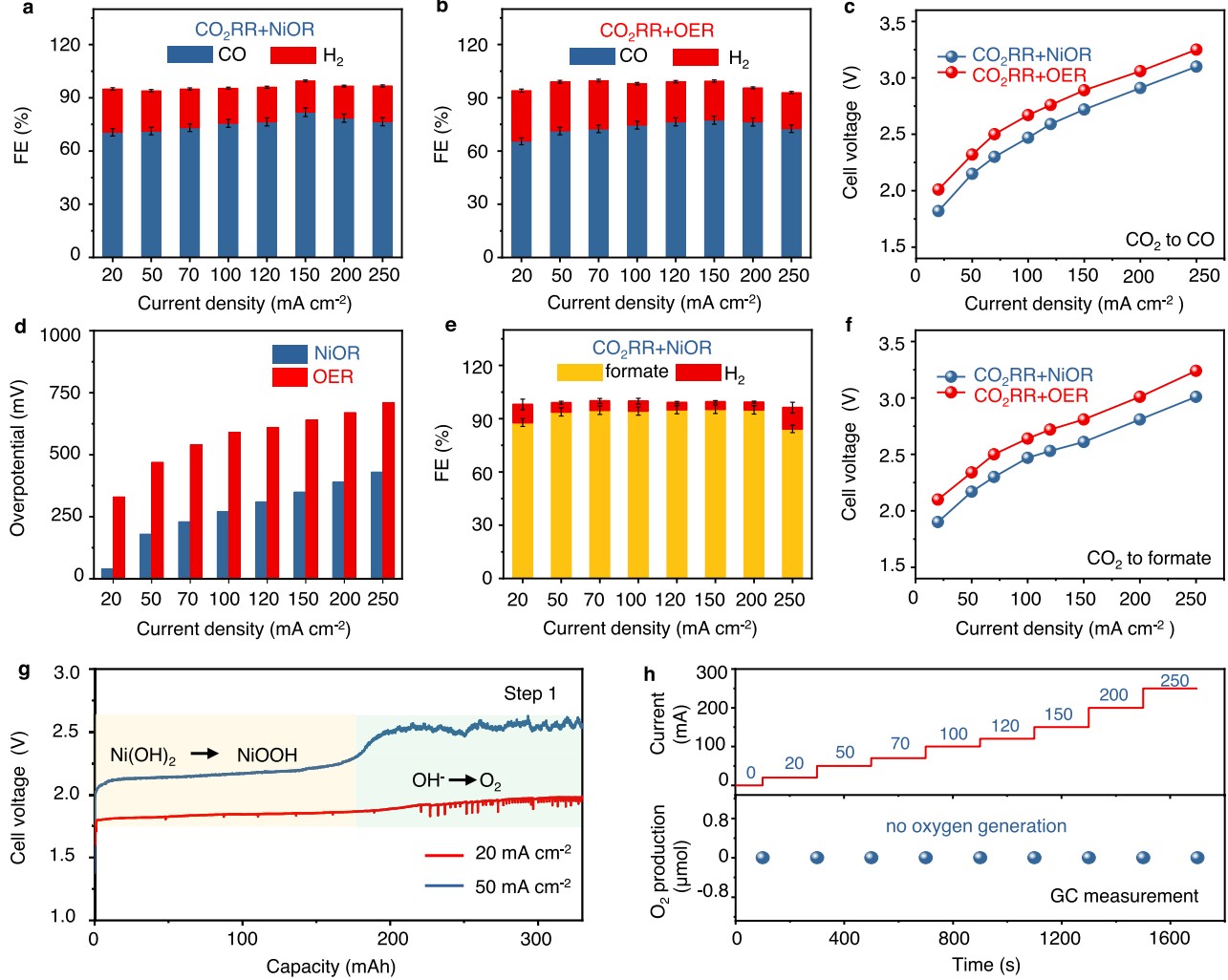

**Fig. 4 | Cell performance in CO₂RR.** Faradaic efficiencies of CO production in **a** CO₂RR + NiOR and **b** CO₂RR + OER; **c** Polarization curve comparison of CO₂RR + NiOR and CO₂RR + OER for CO production; **d** Anode overpotentials of NiOR and OER during CO₂RR at different current densities; **e** Faradaic efficiencies of formate production in CO₂RR + NiOR; **f** Polarization curve comparison of CO₂RR + NiOR and CO₂RR + OER for formate production; **g** Chronopotentiometry curves of Step 1 (CO₂RR+NiOR) for CO production at 20 and 50 mA cm⁻²; **h** O₂ production at the anode in Step 1 at different current densities. The error bar represents standard deviation from three independent measurements. Source data are provided as a Source Data file.

during the heat treatment when fabricating the electrode (see Supplementary Fig. 19).

We then applied both Zn- and Bi₂O₃-based GDE in our cell to examine the H₂-integrated CO₂RR. In Step 1 for CO₂ reduction to CO, the FE of CO is comparable with that obtained in the H-cell, implying the effectiveness of Zn based GDE with a gradient functional layer (see Fig. 4a). The maximum CO FE reached 81.9% at 150 mA cm⁻², more details including CO FE, cell voltages, cathodic potentials, and anodic potentials at 20-250 mA cm⁻² are listed in Supplementary Table 7. Note that the sum of FEs for both CO and H₂ is not 100% as the soluble products in the flowing electrolyte are not analyzed.

In the control experiment where NiOR is replaced by OER at the anode using Co₃O₄ catalyst, the observed CO FE is similar with our Step 1 process at all current densities (see Fig. 4b and Supplementary Table 8). This result indicates that the substitution of OER by NiOR has no influence on the cathodic reaction, which is further supported by the essentially identical overpotentials monitored at the cathode (see Supplementary Fig. 20). Importantly, the cell voltages of Step 1 are 0.15 ~ 0.20 V lower than the conventional processes at all current densities (see Fig. 4c, Supplementary Figs. 21 and 22). By respectively monitoring the cathodic and anodic overpotentials, we conclude that this voltage decrease is ascribed to the replacement of OER by NiOR as

shown in Fig. 4d. For example, the overpotential of OER is 330 mV at 20 mA cm⁻², yet is only 40 mV for NiOR. Indeed, OER is thermodynamically more favorable with lower standard reduction potentials (0.40 vs. 0.49 V for NiOR). But the coupled transfer of 4 electrons and 4 protons makes OER kinetically sluggish. The evolution of molecular oxygen gas may also cause additional overpotential loss due to mass transport. Conversely, NiOR is a single-electron/proton transfer process without gas evolution.

Similarly, in Step 1 for CO₂RR to formate using Bi₂O₃ based GDE, the maximum formate FE tops ~95.3% at 150 mA cm⁻² (see Fig. 4e, Supplementary Fig. 23 and Supplementary Table 9 for details). The cell voltages are 0.17 ~ 0.23 V lower than the conventional processes at the same current densities (see Fig. 4f). This difference is also due to the kinetic advantage of NiOR as discussed above (see Supplementary Fig. 23, Supplementary Tables 9 and 10). We thus infer that our cell architecture, together with the paired NiOR, is flexible and effective in enhancing voltage efficiency for reactions toward both gaseous (CO) and soluble (formate) products. The steady-state study for an extended period is also performed. Figure 4g shows the example for CO generation. At 20 mA cm⁻², the average electrolysis voltage is only 1.85 V, corresponding to a voltage efficiency of 78.4% (see Supplementary Note 1). At 50 mA cm⁻², this voltage increases to 2.15 V;

the full conversion of $Ni(OH)_2$ is indicated by a sharp voltage rise to 2.52 V, implying the start of OER. Continuous operation of $CO_2RR$ must incorporate $H_2$ oxidation in Step 2 which is discussed below.

Albeit that $CO_2$ crossover from cathode to anode remains possible in the new cell during $CO_2RR$, the readily formed $CO_2$ gas can be retrieved easily at the anode side if no oxygen forms during NiOR. We thus used online gas chromatography (GC) to monitor the anode affluent in which no oxygen has been detected (see Fig. 4h). Note that to facilitate the gas collection at the anode while maintaining electrolyte flow at the cathode, we slightly modified the cell by placing an anion-exchange membrane to separate the anode and cathode compartment (see details in the SI and Supplementary Fig. 24). This modification also suggests that our cell is flexible with both membrane-free and membrane-based configurations. The absence of OER at the anode is also supported by the in-situ differential electrochemical mass spectrometry (DEMS) measurement in which $Ni(OH)_2$ was used as the working electrode biased at various constant current densities (see details in the SI and Supplementary Fig. 25)[46]. No $O_2$ signal (m/z = 32) is detected in the current density ranging from 0 to 70 mA cm$^{-2}$ (280 mA cm$^{-2}$ equivalent in the cell) for pristine $Ni(OH)_2$ electrode. Further study indicates that the oxygen formation is also affected by the DoC of $Ni(OH)_2$ electrode. When DoC is below 20.8%, oxygen is undetectable below 150 mA cm$^{-2}$ equivalent (see Supplementary Figs. 26 and 27). When the applied current density is below 40 mA cm$^{-2}$ equivalent, no OER occurs even when DoC reaches >90%.

After full conversion of $Ni(OH)_2$ to NiOOH in Step 1, hydrogen utilization and energy harvesting is then conducted in Step 2. As a proof-of-concept, we used Pt/C GDE for HOR without developing a new catalyst. The electrolyte is switched to 6 M KOH to minimize polarization losses. The maximum power density in Fig. 5a reaches 221 mW cm$^{-2}$ which is comparable with the state-of-the-art Ni-$H_2$ and Zn-air batteries[47,48] (see Supplementary Fig. 28). In the control experiment, we tested $CO_2RR$ with HOR simultaneously occurring at the anode without using $Ni(OH)_2$/NiOOH mediator (see the cell structure in Supplementary Fig. 29). The polarization curve from the very first scan indeed shows substantially lower operating voltage. Yet, in the 2nd and 3rd run, performance deteriorates as the voltage sharply increases to values that are similar to the conventional $CO_2RR$ (Fig. 5b). This phenomenon is aligned with the steady-state electrolysis in which a sudden voltage or current density degradation is observed after a period of time (see Supplementary Fig. 30). We analyzed the spent HOR electrode using the Fourier transformed infrared spectroscopy (FTIR) as a function of electrolysis time. A wide band at ca. 2100–2000 cm$^{-1}$, which corresponds to atop adsorbed CO species on Pt[49], starts to appear after ~300 s (see the inset of Fig. 5b). We thus ascribe the degradation to CO poisoning of Pt electrode. Additionally, formate, methanol and ethanol could deactivate Pt too as shown in Supplementary Fig. 31. These $CO_2RR$ products can migrate through the liquid electrolyte or AEM, occupying the active sites for hydrogen oxidation[50]. Therefore, $Ni(OH)_2$/NiOOH mediator is indeed necessary and effective in tackling both carbon loss and HOR catalyst poisoning.

The steady-state study of Step 2 at galvanostatic conditions is shown in Fig. 5c. At 20 and 50 mA cm$^{-2}$, the cell voltage is ~1.28 V, comparable with the state-of-the-art Ni-$H_2$ batteries. The voltage efficiency is higher than 95%, which outperforms that of the proton-exchange membrane fuel cell (PEMFC) in our own test and in the literature (see Supplementary Fig. 32 and Supplementary Table 13)[51]. Combined with the data in Step 1, we find the Coulombic efficiency of the mediator is >99%. Figure 5d shows the periodical swap between Step 1 and Step 2 at different current densities. While relatively stable voltages are recorded from 20 to 150 mA cm$^{-2}$, the voltage efficiencies, for both CO and formate generation in Fig. 5e, decrease when the current density rises. In the multi-swap test, we select 50 mA cm$^{-2}$ as the benchmark for both Step 1 and Step 2. In the $CO_2$-to-CO conversion shown in Fig. 5f, the operation time of each cycle is set at 2000 s, a

shallow charge-discharge profile can extend the lifespan of typical Ni electrode to tens of thousands of cycles[44,47]. After 10 full cycles, no degradation is observed and the CO FE remains at $72 \pm 1\%$. In the 100 h longevity test for both $CO_2$-to-CO and $CO_2$-to-formate conversion as shown in Supplementary Figs. 33 and 34. The voltage degradation remains trivial, which only comes from Step 1 (0.11 V and 0.12 V for CO and formate generations, respectively). The spent Zn and $Bi_2O_3$ GDEs show unchanged microstructures in the SEM. In particular, the gradient functional layer of the Zn GDE remains intact, no cracks or delamination is seen in SEM images shown in Supplementary Figs. 35 and 36. No CO contaminant is detected on Pt/C GDE after longevity test (see Supplementary Fig. 37).

Energy efficiency and techno-economic analyses are finally carried out in comparison with the conventional $CO_2RR$. We first used "operating voltage" as the descriptor which directly reflect the energy consumption at a fixed current, $H_2$-integrated $CO_2RR$ shows 0.87 V and 0.89 V cell voltage at 50 mA cm$^{-2}$ (defined as the voltage difference between Step 1 and Step 2) for CO and formate generations, respectively. They are significantly lower than the conventional counterparts and the state-of-the-art in the literature (with and without paired oxidation reactions, see Fig. 5g and Supplementary Table 12). To evaluate the energy efficiency of $H_2$-integrated $CO_2RR$ where the upstream hydrogen generation should be included too, we constructed an AWE and an electrode-supported SOEC (see the SI for all technical details, Supplementary Figs. 38–40) which are representative approaches of today's green hydrogen production. Their polarization curves for water electrolysis are shown in Supplementary Figs. 38 and 40. We also select 50 mA cm$^{-2}$ as the benchmark current density at which SOEC and AWE deliver cell voltages of 0.94 and 1.43 V, respectively. Thus, the equivalent operating voltages of $H_2$-integrated $CO_2RR$ + SOEC and $H_2$-integrated $CO_2RR$ + AWE are 1.81 and 2.30 V for $CO_2RR$-to-CO conversion, respectively, both of which are lower than that of conventional $CO_2RR$ (2.32 V). A detailed half-reaction overpotential ($\eta$) analysis is shown in Fig. 5h and Supplementary Table 11. Interestingly, OER alone in conventional $CO_2RR$ reaches $\eta = 0.47$ V, much higher than the sum of all losses in SOEC, Ni mediator and HOR in Step 2 in the $H_2$-integrated $CO_2RR$ + SOEC ($\eta_{sum} = 0.26$ V). It is also higher than the $\eta_{sum}$ for $H_2$-integrated $CO_2RR$ + AWE (0.44 V). Same conclusion is also drawn in the $CO_2RR$-to-formate conversion. These results imply the kinetic benefits of transferring OER in $CO_2RR$ to a water electrolyzer.

We also use energy consumption (GJ) per tonne of product as the performance descriptor (see Fig. 5i and Supplementary Note 2). Interestingly, the $H_2$-integrated $CO_2RR$ coupled with water electrolysis (SOEC or AWE), despite of the higher level of system complexity, demonstrates up to 23% reduction in energy consumption. For instance, conventional $CO_2RR$ requires 22.4 GJ to produce one tonne of CO whereas $H_2$-integrated $CO_2RR$ + AWE needs 22.2 GJ; the SOEC-involved system is even more advantageous (17.5 GJ). When the anodic $CO_2$ recovery energy is considered in neutral conditions, both AWE- and SOEC-coupled $H_2$-integrated $CO_2RR$ demonstrate ample efficiency advantages, showing 27% ~ 42% decrease in energy consumption. This conclusion is also verified at various operating conditions (see Supplementary Tables 14–19). Indeed, direct $CO_2$-to-CO conversion in SOEC consumes even less energy (e.g., ~13.5 GJ[52]), but the low-temperature $CO_2$ electrolysis enjoys the capability of generating various C1–C3 products. In the preliminary techno-economic analysis, we use an established model by considering a pilot plant producing 100 tonne of CO per day[53]. Not surprisingly, the materials expense of our single $H_2$-integrated $CO_2RR$ cell is higher than the conventional counterpart due to the use of mediator and the incorporation of an advanced water electrolyzer (see Supplementary Note 3 and Supplementary Fig. 41). Such additional capital expenditure can be compensated since the plant can behave as a battery energy storage system and can produce hydrogen too, offering versatile solutions for the market.

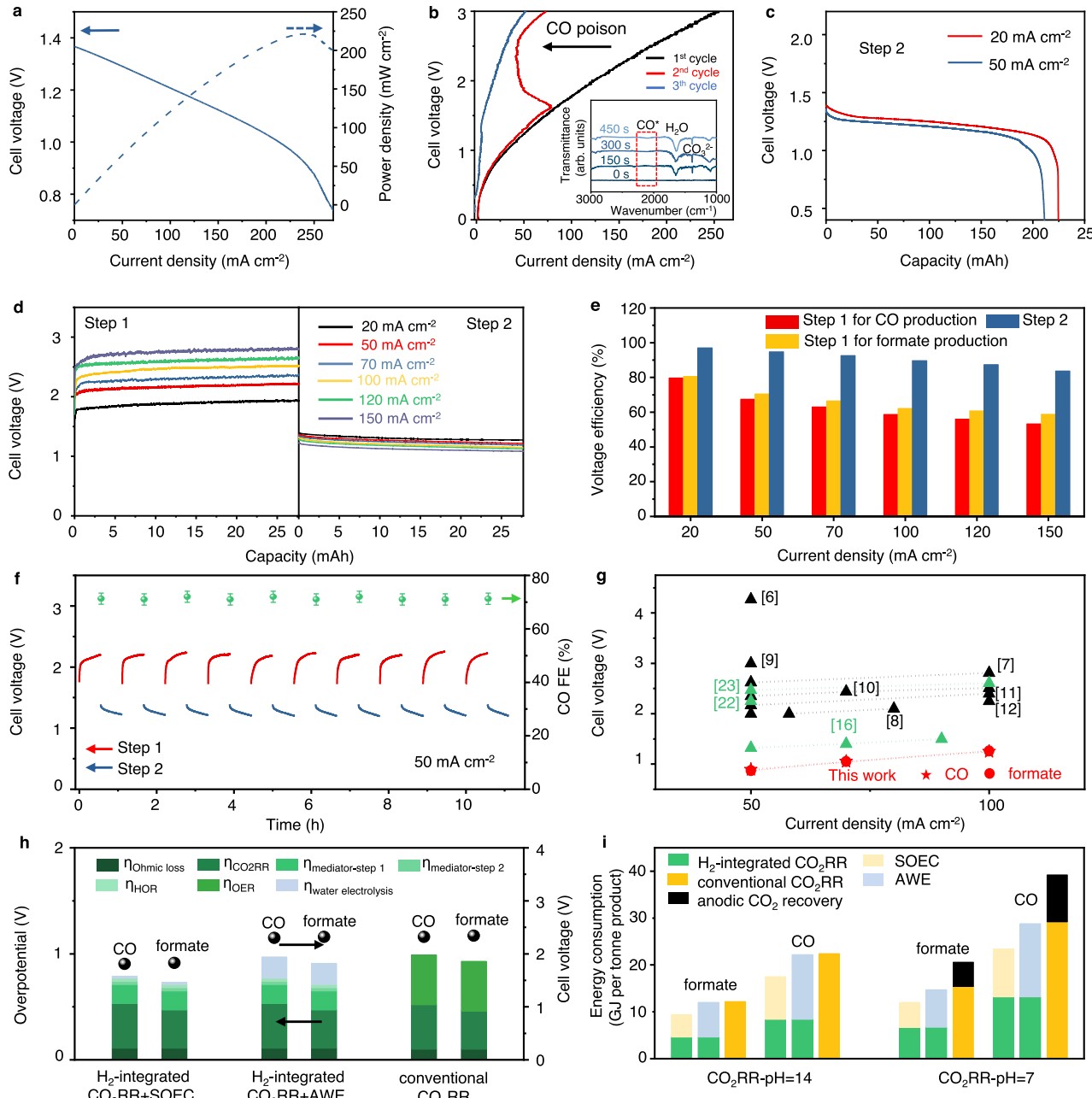

**Fig. 5 | Cell performance in hydrogen conversion and the overall performance assessment. a** Polarization and power density curves of Step 2; **b** Polarization curves of CO$_2$RR directly coupling anodic HOR, the inset shows the FTIR spectroscopy of Pt/C GDE during the HOR process; **c** Chronopotentiometry curves of Step 2 at 20 and 50 mA cm$^{-2}$; **d** Swap between Step 1 and Step 2 for CO generation; **e** Voltage efficiency of Step 1 and Step 2; **f** Multi-swap test between Step 1 and Step 2 at 50 mA cm$^{-2}$ for CO generation; **g** Comparison of operating voltages of

H$_2$-integrated CO$_2$RR and state-of-the-art CO$_2$RR in the literature with (green) and without (black) paired electrooxidations from refs. 6–12,16,22,23; **h** Contributions of polarization losses in H$_2$-integrated CO$_2$RR coupled with water electrolyzers and in conventional CO$_2$RR at 50 mA cm$^{-2}$; **i** Comparison of energy consumptions between H$_2$-integrated CO$_2$RR coupled with water electrolyzer and conventional CO$_2$RR at 50 mA cm$^{-2}$, assuming CO$_2$ recovery costs 4 GJ per tonne of CO$_2$ [20]. Source data are provided as a Source Data file.

## Discussion

We have successfully developed a single electrochemical CO$_2$RR cell pairing H$_2$ oxidation at the anode. The operating voltage of the cell is substantially decreased to <0.9 V at 50 mA cm$^{-2}$. The use of a Ni(OH)$_2$/NiOOH mediator to decouple the electrode reaction effectively mitigates HOR catalyst poisoning while prohibiting anodic carbon loss. Even by including the energy input of hydrogen generation, we found the H$_2$-integrated CO$_2$RR can cut the total energy consumption by 42% thanks to the transferred OER to a water electrolyzer with favored reaction conditions. We

envisage that the direct coupling of CO$_2$RR and HOR without using a redox mediator is more promising which can further cut the energy input by 2.0 GJ per tonne of CO. Thus, finding effective approach of tackling CO$_2$ crossover in neutral conditions and developing poisoning-resistant HOR catalyst becomes critical. While this work demonstrates the opportunity of combining CO$_2$RR with grid-scale energy storage, it might also inspire the community to consider the integration of various emerging energy conversion and storage approaches with the aim of maximizing the energy efficiency.

## Methods

### Preparation procedures of electrodes

**Preparation of Zn-Cu electrode.** Zn nanosheet on Cu foam was prepared by electrodeposition. The aqueous electrolyte solution contained 1.5 M $(NH_4)_2SO_4$ and 0.1 M $ZnSO_4 \cdot 7H_2O$. Before electrodeposition, Cu foam was washed by diluted HCl solution for 5 min, and sequentially rinsed by deionized (DI) water for three times. The deposition was performed at a geometric current density of 4 mA cm$^{-2}$ for different periods of time. To prepare the foam-based GDE, we first immersed the cleaned Zn-Cu foam ($2 \times 2$ cm$^2$) in 10 wt.% plytetrafluoroethylene (PTFE) dispersion (Shanghai Aladdin Biochemical) for 30 s to form a hydrophobic layer on the surface which facilitated the fabrication of the gradient functional layer in the subsequent steps. The carbon-PTFE suspension was prepared by mixing 0.1 g carbon black with different volume of PTFE dispersion via sonication. The carbon ratio to PTFE was 80 wt.%, 50 wt.%, and 20 wt.% (denoted as C-PTFE-80, C-PTFE-50, C-PTFE-20, respectively). To prepare the gradient functional layer, we used a "layer-by-layer" technique via drop-casting 0.5 mL C-PTFE-80 dispersion first on the Zn-Cu foam. The as-prepared electrode was dried in air and then heat-treated in N$_2$ at 350 °C for 30 min. Then, C-PTFE-50 and C-PTFE-20 layer was sequentially deposited using the identical method. In the final structure, each layer was ca. 50–75 μm thick.

**Preparation of Bi$_2$O$_3$ electrode.** The porous Bi$_2$O$_3$ nanosphere was prepared by a hydrothermal synthesis using a template method. To obtain the carbon template, 60 mL of 0.55 M glucose solution was added and sealed in a 100 mL Teflon-lined autoclave, followed by a hydrothermal reaction at 200 °C for 4 h. The retrieved solid was sequentially washed by DI water and ethanol, followed by air-drying at 105 °C for 12 h, to form the template. Then, 0.3 g template and 7.28 g Bi(NO$_3$)$_3$·5H$_2$O were dispersed in 60 mL DI water under sonication. The suspension was sealed in autoclave and heat-treated at 160 °C for 6 h. The retrieved solid was washed by DI water for several times and then air-dried at 105 °C for 12 h. Bi$_2$O$_3$ nanospheres were obtained after calcining the powder at 400 °C for 1 h. To prepare the GDE, 10 mg Bi$_2$O$_3$ powders, 5 mg carbon black and 100 μL 5 wt% Nafion solution were mixed in a solution containing 1 mL ethanol and 1 mL DI water. 210 μL of catalyst ink was infiltrated into the carbon matrix (Sigracet 29 BC, $1 \times 1$ cm$^2$) to form the catalyst layer of a GDE.

**Preparation of Co$_3$O$_4$ electrode.** To prepare the Co$_3$O$_4$ electrode for OER, we used the solvothermal method to grow Co$_3$O$_4$ nanostructures on Ni foam. The solution contained 2.7 mmol Co(NO$_3$)$_2$·6H$_2$O, 2.7 mmol NH$_4$F, 13.5 mmol (NH$_2$)$_2$CO and 30 mL DI water. Then, the acid-washed and cleaned Ni foam ($2 \times 2$ cm$^2$), together with the solution, were sealed in a 50 mL Teflon-lined autoclave. The autoclave was heated up in an oven and maintained at the temperature of 120 °C for 6 h. The obtained Ni foam was washed by deionized water for several times and then dried at 50 °C. The dried electrode was finally calcined at 300 °C to convert the deposited Co species into Co$_3$O$_4$. The sample was donated as Co$_3$O$_4$-Ni.

### Cell assembly

Other than specified, the custom-built cell comprised three electrodes: a GDE for CO$_2$RR ($1 \times 1$ cm$^2$), a Ni(OH)$_2$/NiOOH mediator supported on Ni foam ($2 \times 2$ cm$^2$), and a Pt/C GDE ($1 \times 1$ cm, Pt loading: 0.1 mg cm$^{-2}$) for HOR. A 1.5 cm thick polyether ether ketone (PEEK) frame was placed between the CO$_2$RR GDE and the NiOOH/Ni(OH)$_2$ mediator, where a Hg/HgO reference electrode (RE) was placed. The mediator and HOR GDE were separated by a 130-μm-thick porous separator ($3 \times 3$ cm$^2$, wet-laid nonwoven fabric by polyolefin). The separator was soaked in 1 M KOH for 24 h before use. The schematics and photograph of the cell are shown in the Supplementary Information.

### Materials characterizations

The XRD patterns were recorded by the Rigaku Smartlab X-ray diffractometer using Cu Kα radiation at 40 kV and 44 mA. The SEM images were obtained using the Hitachi-S4800. O$_2$ production was monitored via the differential electrochemical mass spectrometry (DEMS) system (Shanghai LingLu Instrument Corp., Ltd., China) containing a PrismaPlus quadrupole mass spectrometer from Pfeiffer Vacuum and a custom-built Swagelok cell. Gas product was quantified using a gas chromatography (GC-2014C, Shimadzu, Kyoto, Japan) equipped with a thermal conductivity detector and two flame ionization detectors. Liquid product was quantified by a 600 MHz nuclear magnetic resonance spectrometer (NMR, AVANCE NEO 600). The specific surface area was measured with a Micromeritics ASAP-2020 instrument and analyzed by the Brunauer-Emmett-Teller (BET) method. The Fourier transform infrared (FTIR) spectroscopy was performed on a Nicolet iS10 FT-IR spectrometer.

### Electrochemical measurement

**Three-electrode test.** Other than specified, all electrochemical measurements were conducted using a CHI 660E electrochemical workstation (CH Instrument Inc.). A classic H-type cell was used, which was consisted of a 25 mL cathodic compartment and a 25 mL anodic compartment separated by a 183-μm-thick proton exchange membrane (PEM, Dupont N117). Before test, the PEM was first immersed into a 5 wt.% H$_2$O$_2$ solution at 80 °C for 1 h, followed by soaking in DI water for 0.5 h. Then, it was boiled in a 5 wt.% H$_2$SO$_4$ solution at 80 °C for 1 h, followed by soaking in DI water for 0.5 h. Platinum plate ($1 \times 1$ cm$^2$) and saturated calomel electrode (SCE) were used as the counter and reference electrode, respectively. All reported potentials were converted to the reversible hydrogen electrode (RHE) using the equation below:

$$E_{RHE} = E_{SCE} + 0.059 \times pH + 0.242 \qquad (9)$$

Post iR-compensation was applied at 85% value of the solution resistance which was obtained at open circuit potential using EIS. Before CO$_2$RR measurement, CO$_2$ was purged into the 0.1 M KHCO$_3$ aqueous solution for 30 min. During the CO$_2$RR test, CO$_2$ was purged into the cathodic compartment, the measured outlet flow rate was -27 mL min$^{-1}$. EIS was recorded at frequencies ranging from 10$^5$ Hz to 0.01 Hz. ECSA was estimated by measuring an electric double layer capacitance from the scan-rate-dependent CV at various scan rates (20, 40, 60, 80, 100 and 120 mV s$^{-1}$) in an Ar-saturated 0.5 M Na$_2$SO$_4$ electrolyte. The potential window was selected from −0.57 to −0.46 V vs. SCE. The capacitance (C$_{dl}$) was estimated by plotting the difference of charging current density versus the scan rate.

Both NiOR and NiRR of Ni(OH)$_2$/NiOOH electrode, as well as the OER of Co$_3$O$_4$ on Ni foam ($1 \times 1$ cm$^2$) were studied in a three-electrode system. 1 M KOH solution was used as the electrolyte, a platinum plate ($1 \times 1$ cm$^2$) was used as the counter electrode, and a Hg/HgO electrode was used as the reference electrode. LSV and CV was performed at a scan rate of 1 mV s$^{-1}$. The Tafel slope was calculated using the Tafel equation:

$$\eta = b\log_{(j)} + a \qquad (10)$$

where $\eta$, $b$, and $j$ represent the overpotential, Tafel slope, and current density, respectively. All reported potentials in this section were relative to the RHE using the equation below:

$$E_{RHE} = E_{Hg/HgO} + 0.059 \times pH + 0.098 \qquad (11)$$

Post iR-compensation was applied at 85% value of the solution resistance which was obtained at the open circuit potential using EIS.

EIS was carried out at frequencies ranging from $10^5$ Hz to 0.01 Hz. The ohmic resistance was not included in the reported Nyquist plots.

**Full cell test.** In Step 1, $CO_2$ gas was fed to the cathode with a flow rate of ~35 mL min$^{-1}$ and 1 M KOH or 1 M KHCO$_3$ were supplied to the cell with a flow rate of 20 mL min$^{-1}$. The polarization curve was obtained at the steady state by biasing the cell at the specific current density for 20 s before recording the cell voltage. No iR compensation was applied in all full cell test. FE of any gas product was calculated by using the following equation:

$$FE(\%) = \frac{F \times z \times v \times C_j}{I_{overall} \times V_m} \quad (12)$$

where $F$ is the Faraday constant (96485 C mol$^{-1}$), $z$ is the number of electrons transferred to form the CO$_2$RR product j, $v$ is the flow rate of supplied gas, $C_j$ is the detected concentration of gas product, $I_{overall}$ is the overall current, and $V_m$ is the molar volume of gas (22.4 L mol$^{-1}$). FE of liquid products was measured by NMR spectroscopy. The NMR tube contained 0.1 mL of the collected electrolyte solution, 0.1 µL of DMSO as the internal standard and 0.4 mL D$_2$O. The liquid products were quantified by NMR spectroscopy. According to the NMR spectroscopy, the concentration of liquid product could be obtained through the following equation:

$$C_j = \frac{V_i \times \rho_i}{M_i} \times \frac{S_l}{S_i} \times \frac{m_i}{m_l} \times \frac{1}{v_e} \quad (13)$$

where $v_e$ is the volume of the electrolyte in the NMR tube; $\frac{S_l}{S_i}$ is the ratio of the area of the liquid product peaks to the area of the internal standard peak; $\frac{m_i}{m_l}$ is the ratio of the number of the certain protons in the internal standard to that in the liquid product molecules ($m_i$ is 6 from two methyl groups for DMSO; $m_l$ is 1 for formate); $\rho_i$ is the density of internal standard; $V_i$ is the volume of the internal standard solution in the NMR tube; $M_i$ is the molar mass of the internal standard ($M_i$ is 78.13 g mol$^{-1}$ for DMSO). The FE of liquid product was calculated using the following equation:

$$FE(\%) = \frac{F \times z \times v_l \times C_j}{I_{overall}} \quad (14)$$

where $v_l$ is the flow rate of the electrolyte. In Step 2, the electrolyte was switched to 6 M KOH and H$_2$ was fed to the anode with a flow rate of ~10 mL min$^{-1}$.

**Poisoning of Pt/C electrode in HOR.** The HOR performance of Pt/C was evaluated in a three-electrode system using a glassy carbon rotating-disk electrode (RDE, Pine Instrument, disk area of 0.196 cm$^2$). 1 M KOH solution with/without 0.1 M methanol, 0.1 M ethanol or 0.1 M formate was used as the electrolyte, respectively. A Pt wire was used as the counter electrode, and a Hg/HgO electrode was used as the reference electrode. The catalyst ink contained 4 mg 20 wt.% Pt/C powder, 950 µL ethanol and 50 µL Nafion solution (5 wt%), and was prepared by ultrasonication. Then, 10 µL catalyst ink was drop-casted onto the glassy carbon electrode, resulting in a catalyst loading of ~0.2 mg cm$^{-2}$. Before HOR measurements, the electrolyte was bubbled with high-purity H$_2$ gas for 30 min. Voltammograms were collected at a scan rate of 5 mV s$^{-1}$ at 1600 rpm.

**Fabrication and testing of PEMFC**
The PEMFC comprises a membrane electrode assembly (MEA) and gas diffusion layers (GDL) on both sides. Commercial Pt/C (57.7 wt% Pt, TKK), isopropanol, and 5 wt% Nafion solution (DuPont, USA) were ultrasonically mixed for 1 h. The resulting ink-like slurry was then sprayed onto the opposite sides of a pretreated sulfonic acid resin membrane (Dongyue, DF260) as the cathode and anode, respectively. The Pt loadings were 0.5 mg cm$^{-2}$ and 0.1 mg cm$^{-2}$ for the cathode and anode, respectively. The fuel cell was tested with a backpressure of 30 psi at both electrode chamber, and the test temperature was fixed at 70 °C. Both H$_2$ and O$_2$ (air) were humidified before feeding into the fuel cell.

**Fabrication and testing of AWE**
We assembled a zero-gap AWE to evaluate the performance in water splitting[13]. Briefly, the hydrothermally formed Co- and Ni-hydroxides on Ni foam was phosphidated in the atmosphere of PH$_3$-containing Ar at 300 °C. The obtained Co-Ni phosphide/spinel oxide hybrid nanostructure was super-aerophobic and active for both HER and OER. The electrolyzer comprised two titanium plates as the current collector. Porous polyethersulfone (PES) with a thickness of 0.12 mm was used as the separator. 6 M KOH solution was used as the circulating anolyte and catholyte using a peristaltic pump at 40 rpm. The electrolyte containers and the electrolyzer were submerged in a water bath at 85 °C. The hot water was circulated to avoid overheating of the electrolyzer during operation. No iR compensation was applied.

**Fabrication and testing of SOEC**
We fabricated an electrode supported SOEC to evaluate the high-temperature water splitting[54]. Briefly, 33 wt% YSZ (yttria stabilized zirconia; TZ-8Y, Tosoh Corporation), 45 wt% NiO and 22 wt% corn starch were mixed in ethanol by ball milling. The retrieved powder was pressed into discs of ca. 1.27 cm in diameter and then sintered at 1100 °C for 2 h to obtain strong substrates. We then sequentially applied a functional layer (50 wt% NiO + 50 wt% YSZ, no pore-forming agent) and YSZ electrolyte via spin coating. The half-cell was densified at 1420 °C for 4 h. Finally, the air electrode comprising 50 wt% home-made La$_{0.8}$Sr$_{0.2}$MnO$_{3-x}$ and 50 wt% YSZ was applied on the densified electrolyte. In the test, the SOEC button cell was mounted between a pair of co-axile ZrO$_2$ tube with the fuel electrode sealed by ceramic sealant. The detailed design and photos of the home-built SOEC test setup were shown in Supplementary Fig. 39. The cell was preconditioned in 5% H$_2$ + N$_2$ at 650 °C to fully reduce NiO into Ni. During water electrolysis, 50% H$_2$O (steam) + H$_2$ was fed into the fuel electrode using a mass flow controller and a steam generator. The inlet tube was heated to 105 °C to avoid water condensation. No iR compensation was applied.

## Data availability
Source data of Fig. 2, Fig. 4, and Fig. 5 is provided with this paper. Datasets presented in the Supplementary Information are available from the corresponding author on request. Source data are provided with this paper.

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

## Acknowledgements

We acknowledge the funding through National Natural Science Foundation of China (52272233, to N.Y.), Guangdong Basic and Applied Basic Research Foundation (2023A1515011161, to N.Y.) and the Netherlands Organization for Scientific Research (NWO) Vidi grant (VI.Vidi.192.045, to N.Y.). Y.S. thanks the funding from Shenzhen Science and Technology Program (JCYJ20220530143401002, to Y.S.). The authors would also like to thank the Center for Electron Microscopy at Wuhan University for support on the microstructural characterizations, Large-scale Instrument and Equipment Sharing Foundation of Wuhan University, Dr. Ran Zhang and Dr. Yucheng Liu from the Core Facility of Wuhan University for their assistance with NMR and SEM analysis, and Prof. Xiangheng Xiao for the support in GC analysis.

## Author contributions

X. J. and N. Y. conceived and guided the project. X. J. performed materials synthesis, characterizations, and electrochemical tests. L. K., K. Z. and X. Y. assisted in the cell design and fabrication. H. W. conducted GC analysis. X. C., Y. L. and L. L. performed the AWE test. Y.S. and Z.W. offered materials characterizations and data analysis. D.D. performed the PEMFC test. X. J. and N. Y. wrote the manuscript with input from all authors.

## Competing interests

The authors declare no competing interests.
