## [Peer Review File · Nature Communications]

Reviewers' comments:

Reviewer #1 (Remarks to the Author):

The manuscript by Jiang et al developed an integrated cell device for CO₂ electrolysis toward CO while using hydrogen oxidation as the counter reaction. To resolve an anode poison issue, a Ni(OH)₂ solid-state redox mediator was used. It is a systematic study with merits on renewable energy storage and chemical production toward industrial scale. However, the novelty of the integration and the claimed scale fall short of the expectation and the journal standard.

1. Ni(OH)₂ redox is a one-electron process with lower barrier than the water oxidation which is 4-electron process. It certainly has kinetic advantage. However, the thermodynamic advantage is poorly justified or maybe misinterpreted. CO₂ electrolysis is an energy-chemical conversion as a mean for energy storage in addition to manufacturing useful chemicals. For energy storage, the thermodynamic argument is not valid since large energy requirement does not mean disadvantage. Starting from H₂ beats the purpose of energy storage in the first place, at least if the product is CO which is not a better carrier than H₂ for energy storage.

2. It states "which delivered 50 mA cm⁻² at a voltage of

26 only 0.95 V, consuming ~60% less electrical energy than the conventional counterpart.". The device is not generating useful potential. It is still consuming extra energy. 60% less is not meaningful since the starting point is H₂ vs H₂O.

3. On another aspect, the "hydrogen-electricity" conversion is claimed to offer ~20% voltage efficiency increase. This is poorly defined in terms of efficiency. The device is not a fuel cell.

4. The use of Ni(OH)₂ mediator is good for alleviating the anode poison issue. There are also known materials in literature to have less poison problems for H₂ oxidation electrode, such Pt/Ru catalysts. This makes the integration less necessary and might complicate the cell design and application.

5. Instead of thermodynamic and kinetic discussions, economic analysis would be necessary to make the device integration appealing, if such integration is advantageous.

Reviewer #2 (Remarks to the Author):

In this manuscript, the authors report paired electrolysis of CO₂RR at the cathode and hydrogen oxidation (HOR) at the anode to decrease the energy demand of electrolyzers. The data presented is mostly based on comparison of overpotentials and cell performance, whereas the rationale behind the study was also largely kinetic (sluggish kinetics of OER). I think kinetic experiments are invaluable to support the presented claims, and without such kinetic experiments (Tafel plots, EIS, etc) I do not think

this work suits the scope of Nature Communications. If these experiments can be included in a revised version, and they support the presented claims, I think this work could be suitable. I look forward to a revised version of the manuscript.

Reviewer #3 (Remarks to the Author):

The manuscript submitted to Nature Communications by Jiang et al. reported on a HOR coupled CO₂R system using Ni(OH)₂/NiOOH as a mediator to prevent carbon loss and HOR catalyst poisoning. In this system, the cell voltage was reported to be 0.95 V at 50 mA cm⁻², which is a significant improvement over the traditional CO₂ reduction reaction. The idea of using a Ni(OH)₂/NiOOH redox mediator has merit. However, several details must be addressed:

(1) In the long-term test (Fig. 5d), the CO₂R duty cycle should be detailed and further explored. For instance, is it possible to alter the polarization rate of Step 2 to increase the CO₂R operation time?

(2) It is noted that even under 250 mA cm⁻² (Fig. 4f), there is no O₂ product. But the demonstrated operation time of high current is quite short. At 50 mA cm⁻², the Step 1 (CO₂R) can be operated smoothly for ~ 3h before OER starts (Fig. 4e). What about running Step 1 at a larger current density > 100 mA cm⁻²? How long can it stay stable without OER? There might be a trade-off between the polarization current density and the duty time. It would be great if these two factors of Step 1 can be further optimized to achieve an industrial-relevant level.

(3) The authors target enhancing the total CO₂R energy efficiency. The overall energy efficiency of this system needs to be provided and compared to all relevant alternative approaches. First on an energy basis, is the energy input here – including that required to electrocatalytically produce the H₂ – less than the energy input required to produce the same output product via other means (including SOEC)? Then also cost and carbon comparisons for this case vs. the incumbent.

(4) Some small errors need to be carefully checked. E.g. the figure caption order of Fig. S7.

(5) I'm not sure why there is a H₂ fuel cell in 1a, and the associated narrative. I suppose the concept is to show that OER and ORR are costly in both CO₂ electrolyzers and H₂ fuel cells... but these technologies are not analogous in the way that H₂ electrolyzers and fuel cells are. H₂ makes electricity in the fuel cell case, it would help make carbon-based products more efficiently in the proposed case, and these are very different outcomes.

(6) Throughout the wording and grammar could be improved. I suggest, for instance, changing the title to "Integrating hydrogen conversion in CO₂ electrolysis without oxygen evolution". "The sluggish oxygen reduction reaction was also eliminated" this is of course a big part of the low V achieved, so "also" is not right. "useless" oxygen catalysis. Etc. the writing requires professional proof reading.

(7) Figure quality must be improved to meet the standards and norms of the journal.

In summary, I appreciate the approach and the low-V achieved. The above must be addressed - and the energy cost found to be compelling - for the work to be published in this journal

Comments and revisions of the first reviewer:

Reviewer #1 (Remarks to the Author):

The manuscript by Jiang et al developed an integrated cell device for CO₂ electrolysis toward CO while using hydrogen oxidation as the counter reaction. To resolve an anode poison issue, a Ni(OH)₂ solid-state redox mediator was used. It is a systematic study with merits on renewable energy storage and chemical production toward industrial scale. However, the novelty of the integration and the claimed scale fall short of the expectation and the journal standard.

1. Ni(OH)₂ redox is an one-electron process with lower barrier than the water oxidation which is 4-electron process. It certainly has kinetic advantage. However, the thermodynamic advantage is poorly justified or maybe misinterpreted. CO₂ electrolysis is an energy-chemical conversion as a mean for energy storage in addition to manufacturing useful chemicals. For energy storage, the thermodynamic argument is not valid since large energy requirement does not mean disadvantage. Starting from H₂ beats the purpose of energy storage in the first place, at least if the product is CO which is not a better carrier than H₂ for energy storage.

Response: We agree with the referee that the Ni(OH)₂ redox process has no thermodynamic advantage. It is our fault that the main argument of this work is not well presented in the previous version, all statements that might mislead readers have been removed now. Our ideal is to include “hydrogen cycle” in the CO₂RR for efficiency enhancement as shown in the revised Figure 1a (also see below), the new process eliminates both OER and ORR processes in CO₂RR and in “hydrogen-electricity” conversion process, respectively. The following discussion have been added and highlighted on Page 4.

*“The renewable energy, to power CO₂RR, is intermittent in nature. This means the continuous operation of CO₂ electrolysis requires the integration of reliable and affordable energy storage solutions which are also critical for supporting the worldwide deployment of renewable energy sources. H₂ is often regarded as a clean and flexible energy carrier which can “store” surplus renewable energy via water electrolysis and “release” electricity using fuel cells. Indeed, powering CO₂ electrolysis using electricity generated in hydrogen fuel cells is a good solution in this context as shown in the flowchart of **Figure 1a**. One would note that oxygen reduction reaction (ORR) in fuel cells is the reverse process of OER in CO₂RR, potentially offering opportunities of “cancelling” both half reactions in the cascade process. This is particularly advantageous as both reactions are sluggish 4-electron/proton transfer processes, drastically decreasing the energy efficiency of the respective units.”*

Figure 1. (a) Illustrative comparison of CO₂RR electrolysis powered by renewable energies stored in the form of hydrogen via conventional and H₂-integrated routes.

Regarding the criticism on CO₂-to-CO conversion, we have now developed new electrode and included CO₂-to-formate as another model reaction. These two processes verify that our design is necessary and compatible for both gaseous and soluble product generation with high Faradaic efficiency and low overpotential losses. The key sets of data are attached below for your convenience and included at various places in the manuscript and SI).

“In the CO₂-to-formate model reaction, we also used nanostructured electrocatalyst yet comprising Bi₂O₃. In the hydrothermal synthesis, porous carbon nanospheres were added as the template on which Bi species was deposited. A final calcination removed the carbon template and convert Bi species into Bi₂O₃ porous nanospheres (see the characterizations in Figure S10). This catalyst is highly selective during potentiostatic CO₂ electrolysis to yield formate in the potential range of -0.45 to -1.05 V. The top formate Faradaic efficiency reached 89.0% at -0.65 V as shown in Figure S11. More electrochemical data can be found in the Supplementary Information (Figure S11-S12).”

Figure S10. (a) XRD pattern, (b) SEM image and (c) EDS elemental mappings of nanosphere Bi₂O₃.

Figure S11. Formate FE of Bi₂O₃ at different potentials in a H-cell.

Figure S12. LSV curve of Bi₂O₃ in CO₂-saturated 0.5 M KHCO₃ electrolyte with a scan rate of 5 mV s⁻¹.

“Similarly, in Step 1 for CO₂ reduction to formate using Bi₂O₃ based GDE, the maximum formate FE topped ~95.3% at 150 mA cm⁻² (see **Figures 4e**, S23 and Table S9 for details). The cell voltage was 0.17 ~ 0.23 V lower than the conventional process based on OER at the same current densities (see **Figure 4f**). This difference was also due to the kinetic advantage of NiOR as discussed above (see Figure S23 and Tables S9, S10). We thus concluded that our cell architecture, together with the paired NiOR, is flexible and effective in enhancing voltage efficiency for CO₂RR to both gaseous (CO) and soluble (formate) products.”

Figure 4. (e) Faradaic efficiencies in CO₂RR+NiOR for formate production; (f) comparison of the polarization curves for CO₂RR+NiOR and CO₂RR+OER systems for formate production.

Figure S23. (a) ^1H NMR spectra of the cathode effluent from the cell based on a Bi_2O_3 GDE; (b) Faradaic efficiencies in $\text{CO}_2\text{RR}+\text{OER}$ system for formate production; (c) polarization curves comparison of $\text{CO}_2\text{RR}+\text{NiOR}$ and $\text{CO}_2\text{RR}+\text{OER}$ for formate production; (d) polarization curves of cathode and anode in the combined $\text{CO}_2\text{RR}+\text{OER}$ and $\text{CO}_2\text{RR}+\text{NiOR}$ processes for formate production.

Table S9. The product FE, cell voltage, electrode potentials, and overpotentials (η) in combined $\text{CO}_2\text{RR}+\text{NiOR}$ for formate production.

Current density (mA cm^{-2})	H_2 FE (%)	Formate FE (%)	Cell voltage (V)	Potential _{cathode} (V vs. RHE)	η_{cathode} (mV)	Potential _{anode} (V vs. RHE)	η_{anode} (mV)
20	10.2	87.8	1.90	-0.46	280	1.39	40
50	5.1	93.8	2.17	-0.54	360	1.53	180
70	5.2	94.8	2.30	-0.60	420	1.58	230
100	5.6	94.3	2.47	-0.70	520	1.64	290
120	4.0	95.1	2.53	-0.73	550	1.67	320
150	4.2	95.3	2.61	-0.75	570	1.70	350
200	4.3	95.0	2.81	-0.77	590	1.74	390
250	12.0	84.2	3.01	-0.79	610	1.78	430

Table S10. The product FE, cell voltage, electrode potentials, and overpotentials (η) in combined $\text{CO}_2\text{RR}+\text{OER}$ for formate production.

Current density (mA cm^{-2})	H_2 FE (%)	Formate FE (%)	Cell voltage (V)	Potential _{cathode} (V vs. RHE)	η_{cathode} (mV)	Potential _{anode} (V vs. RHE)	η_{anode} (mV)
20	10.0	87.8	2.10	-0.47	290	1.56	330
50	5.0	92.7	2.34	-0.54	360	1.70	470
70	5.0	93.3	2.50	-0.60	420	1.77	540
100	5.0	93.6	2.64	-0.70	520	1.82	590

120	5.0	94.0	2.72	-0.73	550	1.84	610
150	4.0	95.0	2.81	-0.75	570	1.87	640
200	4.0	93.7	3.01	-0.76	580	1.90	670
250	13.0	83.0	3.24	-0.80	620	1.94	710

“Similar conclusion was also drawn in the CO₂-to-formate conversion that showed up to 50 h of stable operation in Figure S32.”

Figure S32. Longevity test of the system combining periodical swap between Step 1 and Step 2 at 50 mA cm⁻² for formate production.

2. It states "which delivered 50 mA cm⁻² at a voltage of only 0.95 V, consuming ~60% less electrical energy than the conventional counterpart.". The device is not generating useful potential. It is still consuming extra energy. 60% less is not meaningful since the starting point is H₂ vs H₂O.

Response: We understand the concern of the referee and have now rephrased the statement as “This process eliminated both oxygen evolution reaction (OER) in conventional CO₂RR and oxygen reduction reaction (ORR) in fuel cells for “H₂-electricity” conversion, showing up to 40.0% decrease of polarization loss.” Besides, in the discussion section, we have revised the discussions and Figure 5g as shown below and highlighted on Page 20.

“We finally assessed the combined polarization losses at 50 mA cm⁻² during electrolysis as shown in **Figure 5g**. In H₂-integrated CO₂RR, the overall operating voltage (defined as the voltage difference between Step 1 and Step 2) at 50 mA cm⁻² is 0.95 V and 0.97 V for CO and formate generations, respectively. Yet, it reached 2.32 V and 2.34 V in the conventional counterparts (see the comparison with literature in Table S12). A detailed analysis showed that CO₂RR and the redox cycle of Ni electrode were the main overpotential contributors in H₂-integrated CO₂RR to CO, accounting for 0.42 V and 0.23 V losses, respectively. Conversely, OER alone in conventional process contributed 0.47 V overpotential, more than twice higher than that of the Ni electrode. Importantly, when considering conventional CO₂RR powered by stored hydrogen using PEMFC, an application scenario discussed in Figure 1a, we found the total polarization loss was 62.4 ~ 67.1% higher than our process (CO production: 0.85 V vs. 1.38 V; formate production: 0.79 V vs. 1.32 V). The detailed polarization losses of each electrode/process are also listed in Table S13.”

Figure 5g. The contribution of various polarization losses in the H₂-integrated and conventional CO₂RR process at 50 mA cm⁻².

Table S13. Various polarization losses in the H₂-integrated and conventional CO₂RR process at 50 mA cm⁻².

Overpotential (V)	H ₂ -integrated CO ₂ RR to CO	H ₂ -integrated CO ₂ RR to formate	PEMFC	conventional CO ₂ RR to CO	conventional CO ₂ RR to formate
Ohm loss	0.11	0.11	-	0.1	0.1
η _{CO2RR}	0.42	0.36	-	0.42	0.36
η _{AE1}	0.18	0.18	-	-	-
η _{AE2}	0.05	0.05	-	-	-
η _{OER}	-	-	-	0.47	0.47
η _{HOR}	0.09	0.09	-	-	-
η _{PEMFC}	-	-	0.39	-	-

3. On another aspect, the “hydrogen-electricity” conversion is claimed to offer ~20% voltage efficiency increase. This is poorly defined in terms of efficiency. The device is not a fuel cell.

Response: We have now defined the voltage efficiency as shown below and have included the PEMFC test (to compare with our Ni-H₂ battery in Step 2). the statement has been revised (see question 2) and the voltage efficiency is now detailed in Supplementary Note 3 on Page 65 of SI (see below).

“The voltage efficiency of CO₂RR (Step 1 or conventional manner) was calculated as follows:

$$\text{Voltage efficiency}_1 = \frac{E_0}{E_c} \times 100\% \quad (12)$$

where E_c is the measured cell voltage, E_0 is the thermoneutral potential.

The voltage efficiency of “H₂-electricity” conversion (Step 2 or fuel cell) was calculated as follows:

$$\text{Voltage efficiency}_2 = \frac{E_c}{E_0} \times 100\% \quad (13)$$

where E_0 is the Nernst potential. For Step 2 in our cell, E_0 is equal to 1.35 V. And for conventional H₂ fuel cell, E_0 is equal to 1.23V.”

4. The use of Ni(OH)₂ mediator is good for alleviating the anode poison issue. There are also known

materials in literature to have less poison problems for H_2 oxidation electrode, such Pt/Ru catalysts. This makes the integration less necessary and might complicate the cell design and application.

Response: Poison problem might be addressed by developing alternative catalysts, yet CO_2 loss cannot (in fact, this is a serious engineering problem in CO_2RR valorization). The use of the redox process is a simple and effective method as we have shown in the manuscript.

5. Instead of thermodynamic and kinetic discussions, economic analysis would be necessary to make the device integration appealing, if such integration is advantageous.

Response: This is a very good point and we have now carried out the techno-economic analysis to evaluate the materials cost of our H_2 -integrated CO_2RR (Route 1) based on the production of CO in comparison with two alternative approaches (Route 2: conventional CO_2RR flow cell combined with PEMFC; Route 3: solid-oxide electrolysis cells combined PEMFC). The schematic comparison of three routes is shown in Figure S35. Our calculation indicates Route 1 costs ~\$23 and ~\$27 less than Route 2 and Route 3 per kilowatt, respectively. The following discussion and calculation process were added and highlighted in the Page 21 of revised manuscript and Page 60-61 of SI.

“We also carried out a preliminary techno-economic and energy efficiency analysis of CO_2RR coupling H_2 utilization using several representative technologies including flow cells and solid oxide electrolysis cells (SOECs, see all routes in Figure S35). Indeed, our cell is more costly than the conventional CO_2RR flow cells due to the use of mediator and Pt-containing GDE, yet it excludes the integration of PEMFC with expensive membrane and an order of magnitude higher Pt loading, yielding a much lower total cost. In terms of the overall energy conversion efficiency, our cell is comparable with the route involving highly-efficient SOEC while enjoying the capability of generating various C1-C3 products.”

Figure S35. (a) Schematic illustration of Route 1, Route 2 and Route 3 for combining CO_2RR with H_2 -based energy storage; (b) materials cost comparison of three routes per kilowatt; (c) energy efficiency comparison of three routes.

“Supplementary Note 1: Techno-economic analysis

We carried out the techno-economic analysis to evaluate the materials cost of our H_2 -integrated CO_2RR (Route 1) based on the production of CO in comparison with two alternative

approaches (Route 2: conventional CO₂RR flow cell combined with PEMFC; Route 3: solid-oxide electrolysis cells combined PEMFC). The schematic comparison of three routes is shown in Figure S35. Note that only the component of a single cell is considered for more accurate estimation. The materials cost was calculated via the following equation:

$$C_{\text{materials}} = \frac{\sum_n p_n A_n}{P} \quad (7)$$

where P is the operation power of the cell, p_n is unit price of the cell component n , and A_n is the area of each cell component. Details are shown below.

The cost calculation for Route 1:

The total materials cost for Route 1 is ~\$78 kW⁻¹ which comprises of:

(1) Cost of Step 1. The cell components contain Zn-Cu foam GDE and Ni(OH)₂/NiOOH electrode (the unit prices were shown in Table S14). We selected the power at 50 mA cm⁻² (2.15 V, 0.11 W cm⁻²) as a reference. Thus, the calculated cost of conventional CO₂RR cell was ~\$64 kW⁻¹.

(2) Cost of Step 2. The cell component only contain Pt GDE and separator as other components were considered in Step 1, because this process is completed in the same cell with Step 1. The selected power was at 0.12 W cm⁻² (maximum power of Step 2, according to the power density plots of Step 2 in Figure 5a). Thus, the calculated cost for Step 2 was ~\$14 kW⁻¹.

2. The cost calculation for Route 2:

The total materials cost for Route 2 is combined of that from the conventional CO₂RR cell and a PEMFC. The total cost for Route 2 is ~\$101 kW⁻¹ which comprises:

(1) PEMFC. The PEMFC components contain two Pt GDEs, and a piece of Nafion membrane (the unit prices were shown in Table S14). The selected power was at 0.66 W cm⁻² (maximum power of PEMFC, according to the power density plots of H₂-air PEMFC in Figure S31). Thus, the calculated cost was ~\$82 kW⁻¹.

(2) Conventional CO₂ RR cell. The selected power was at 50 mA cm⁻² (2.32V, 0.12 W cm⁻²). The cell components contain Zn-Cu foam GDE and Co₃O₄-Ni electrode (the unit prices were shown in Table S14). Thus, the calculated cost was ~\$19 kW⁻¹.

3. The cost calculation for Route 3:

The total materials cost for Route 3 is combined of that from a solid-oxide electrolysis cell (SOEC) and from a PEMFC. The total cost for Route 3 is ~\$105 kW⁻¹ which comprises of:

(1) PEMFC. The cost of this part is the same as that in Route 2.

(2) SOEC. As the SOEC generally operated in the current range of 0.5-1.5 A cm⁻², the selected power was at 500 mA cm⁻² (1.2 V, 0.6 W cm⁻²)³⁴. The cell components refer to our previous work, which contains cathode electrode (made from NiO, ZrO₂, and corn), ZrO₂ electrolyte, and anode electrode (made from La_{0.8}Sr_{0.2}MnO_{3-x}) (the unit prices were shown in Table S14). Thus, the calculated cost was ~\$23 kW⁻¹.

Table S14. The cost of cell component.

Component	Cost	Units	Source
Zn-Cu foam GDE	12.8	\$ m ⁻²	a
Co ₃ O ₄ electrode	9.8	\$ m ⁻²	a
Ni(OH) ₂ /NiOOH	14.0	\$ m ⁻²	a
Pt GDE	16.0	\$ m ⁻²	a

Separator	2.0	\$ m ⁻²	PNAS 115, 11694-11699 (2018)
Nafion membrane	350	\$ m ⁻²	J. Power Sources 288, 187-198 (2015)
Cathodic electrode of SOEC	81.5	\$ m ⁻²	a
Anodic electrode of SOEC	53.8	\$ m ⁻²	a

a. based on the chemicals/materials cost in the Table S15.

Table S15. The cost of chemicals/materials.

Chemicals/materials	Cost	Units	Source
Carbon black	0.04	\$ kg ⁻¹	a
PTFE	10.3	\$ kg ⁻¹	b
Zn	8.0	\$ kg ⁻¹	c
Cu foam	2.4	\$ m ⁻²	c
Co	75.0	\$ kg ⁻¹	c
Ni foam	6.0	\$ m ⁻²	Nat. Energy 6, 517-528 (2021)
Ni	14.0	\$ kg ⁻¹	c
Pt	32035.0	\$ kg ⁻¹	c
NiO	14.0	\$ kg ⁻¹	c
ZrO ₂	45.8	\$ kg ⁻¹	d
Corn	0.19	\$ kg ⁻¹	e
La _{0.8} Sr _{0.2} MnO _{3-x}	7000.0	\$ kg ⁻¹	f
GDL	0.006	\$ m ⁻²	Renew. Sust. Energ. Rev. 154, 111807 (2022).

a. Taken from online report of “carbon black price trend and forecast” (<https://www.chemanalyst.com/>).

b. Taken from online report of “PTFE prices rising due to improvement in downstream market demand” (<https://www.chemanalyst.com/>).

c. Taken from the report of daily metal prices (<https://www.dailymetalprice.com/>).

d. Taken from online trade market. (https://www.made-in-china.com/products-search/hot-china-products/Zro2_Zirconia_Powder_Price.html)

e. Taken from online trade market. (<https://www.derthickscormaze.com/interesting-about-corn/faq-what-is-the-price-of-corn-per-bushel-today.html>)

f. Taken from online trade market. (<https://www.sigmaaldrich.cn/CN/zh/product/aldrich/704261>)

Figure S31. Polarization curves and power density plots of H₂-O₂ and H₂-air PEMFC.

Comments and revisions of the second reviewer:

In this manuscript, the authors report paired electrolysis of CO₂RR at the cathode and hydrogen oxidation (HOR) at the anode to decrease the energy demand of electrolyzers. The data presented is mostly based on comparison of overpotentials and cell performance, whereas the rationale behind the study was also largely kinetic (sluggish kinetics of OER). I think kinetic experiments are invaluable to support the presented claims, and without such kinetic experiments (Tafel plots, EIS, etc) I do not think this work suits the scope of Nature Communications. If these experiments can be included in a revised version, and they support the presented claims, I think this work could be suitable. I look forward to a revised version of the manuscript.

Response: We agree that adding the kinetic study would be very informative and have now carried out additional work to support our claims (Ni redox cycle is advantageous than oxygen cycle). The following tests, Figures and kinetic analysis are included in the revised manuscript (see Figure 2f-2j, Figure S13, S15-S16, and Table S3-S6; the discussion was highlighted on Page 11-12 of revised manuscript). Besides, we also added additional data on the selectivity of the oxidation reaction at the Ni electrode (OER vs Ni(OH)₂-to-NiOOH) as highlighted on Page 17 of revised manuscript (see Figure S26-S27).

*“In the cyclic voltammogram (CV) shown in Figure S13, the redox peaks, corresponding to the Ni²⁺/Ni³⁺ conversion, was clearly observable. The onset potential of OER was ~ 220 mV higher than that of Ni(OH)₂ oxidation reaction (NiOR). As the Nernst potential of NiOR is higher than that of OER, the lower onset potential suggests the much faster reaction kinetics with lower activation energy barrier than that of OER which is a concerted four-electron-proton transfer process. This conclusion is also valid when comparing with a state-of-the-art OER catalyst, namely, nanostructured Co₃O₄ (detailed materials characterizations and OER performance are shown in Figure S14)⁴⁴⁻⁴⁵, with NiOR showing lower onset potential and Tafel slope (see **Figures 2f** and **2g**).*

*The deeper understanding of the dynamic behaviors of Ni(OH)₂ electrode during NiOR was achieved using electrochemical impedance spectroscopy (EIS). The Nyquist plots, together with the fitting results, in **Figure 2h** reveal charge-transfer processes during NiOR at various potentials. The charge-transfer resistance (R_{ct}) and the Warburg impedance (R_w) decreased progressively when the potential bias increased from the open-circuit potential (OCV) to 1.65 V vs. RHE, implying that the NiOR kinetics was controlled by both charge transfer and mass diffusion. Note that further increasing the potential bias caused the rise of R_{ct} due to the initiation of OER (also see Table S3). The dynamic composition of Ni(OH)₂ electrode also affected reaction kinetics. In the Nyquist plots where the depth-of-charge (DoC) of Ni(OH)₂ varied (see **Figure 2i** and Table S4, changing is defined as Ni(OH)₂-to-NiOOH conversion), we noticed R_{ct} remained essentially identical when DoC was 0%, 30% and 60%. However, it drastically increased from 0.20 to 4.19 Ω when DoC reached 100%, implying that the charge transfer process was then dominated by OER. Compared with the R_{ct} of OER on NiOOH and Co₃O₄, the R_{ct} of NiOR was more than an order of magnitude smaller as shown in **Figure 2j** and Table S5. Besides, we also characterized the reverse process, i.e., NiOOH reduction reaction (NiRR), as shown in Figure S15. It seemed that the diffusion process became the rate-limiting step. When the overpotential increased, R_{ct} remained almost constant while R_w showed substantial rise (see Table S6).*

In the steady-state study using the chronopotentiometric method at a constant applied current of $\pm 10 \text{ mA cm}^{-2}$, stable redox reaction was recorded for both the oxidation and reduction reactions (see Figure S16). The high reversibility of Ni electrode in diluted KOH electrolyte was also reflected by the roughly identical charges (ca. 200 C) stored/released in the two reactions. The calculated Coulombic efficiency was $>99\%$. Before the complete consumption of Ni(OH)_2 in the oxidation cycle, oxygen bubbles were hardly observable, implying that a simultaneous OER was unlikely (cf. the EIS spectra with different DoC). Bubbles started to appear when all Ni(OH)_2 was converted into NiOOH (see the comparison in the inset of Figure S16), which was accompanied by a potential increase from 1.45 to 1.54 V. Based on these results, we confirmed that $\text{Ni(OH)}_2/\text{NiOOH}$ mediator could well perform in 1 M KOH.”

Figure 2. (f) LSV curves of Ni(OH)_2 and Co_3O_4 electrode; (g) the nominal Tafel plots of NiOR and OER; (h) Nyquist plots of Ni(OH)_2 electrode acquired at 1.40, 1.50, 1.65, and 1.80 V vs. RHE; (i) Nyquist plots of $\text{Ni(OH)}_2/\text{NiOOH}$ electrode at different state of charge at 1.55 V vs. RHE in which changing is defined as Ni(OH)_2 -to- NiOOH conversion; (j) Nyquist plots measured at 1.55 V vs. RHE of $\text{Ni(OH)}_2/\text{NiOOH}$ electrode (DoC=0% and =100%) and Co_3O_4 .

Figure S13. CV of the Ni(OH)_2 electrode at scan rate of 1 mV s^{-1} in 1.0 M KOH at 25°C .

Figure S15. Nyquist plots of Ni(OH)₂/NiOOH electrode acquired at 1.25, 1.15, and 1.05 V vs. RHE.

Figure S16. Chronopotentiometric curve of Ni(OH)₂/NiOOH conversion at a constant applied current of 10 mA cm⁻²; inset shows photo profiles of Ni(OH)₂ oxidation and overoxidation of Ni(OH)₂ which leads to OER with the bubble evolution.

Table S3. The fitting results of the EIS plot in Figure 2h.

Potential (V vs. RHE)	R _{ct} (Ω)	R _w (Ω)	C(F)
1.40	0.24	2.90	0.81
1.50	0.21	0.90	0.75
1.65	0.15	0.50	0.75
1.80	0.24	-	0.70

Table S4. The fitting results of the EIS plot in Figure 2i.

DoC (%)	R _{ct} (Ω)	R _w (Ω)	C(F)
0	0.20	0.70	0.75
30	0.22	1.20	0.77
60	0.19	1.40	0.77
100	4.19	-	0.70

Table S5. The fitting results of the EIS plot in Figure 2j.

Reaction	R _{ct} (Ω)	R _w (Ω)	C(F)
NiOR	0.20	0.70	0.75
OER-NiOOH	4.19	-	0.70
OER-Co ₃ O ₄	2.45	-	0.84

Table S6. The fitting results of the EIS plot in Figure S15.

Potential (V vs. RHE)	$R_{ct}(\Omega)$	$R_w(\Omega)$	C(F)
1.25	0.20	1.18	0.68
1.15	0.23	1.90	0.87
1.05	0.25	4.30	0.90

“Further study indicated that the oxygen formation was also affected by DoC of the $Ni(OH)_2$ electrode. When DoC was below 20.8%, no OER was observed at a current density equivalent to 150 mA cm^{-2} of the cell (see Figure S26). When the applied current density is equivalent to 40 mA cm^{-2} and below, no OER was monitored even when DoC reached $>90\%$ (see Figure S27).”

Figure S26. The ionic current response in the DEMS measurement in which $Ni(OH)_2$ was used as the working electrode biased at a constant current density equivalent to 150 mA cm^{-2} of the cell.

Figure S27. The ionic current response in the DEMS measurement in which $Ni(OH)_2$ was used as the working electrode biased at a constant current density equivalent to 40 mA cm^{-2} of the cell.

Comments and revisions of the third reviewer:

The manuscript submitted to Nature Communications by Jiang et al. reported on a HOR coupled CO₂RR system using Ni(OH)₂/NiOOH as a mediator to prevent carbon loss and HOR catalyst poisoning. In this system, the cell voltage was reported to be 0.95 V at 50 mA cm⁻², which is a significant improvement over the traditional CO₂ reduction reaction. The idea of using a Ni(OH)₂/NiOOH redox mediator has merit. However, several details must be addressed:

1. In the long-term test (Fig. 5d), the CO₂RR duty cycle should be detailed and further explored. For instance, is it possible to alter the polarization rate of Step 2 to increase the CO₂RR operation time?

Response: We have now performed additional experiment with Step 1 and Step 2 being carried out at various current densities (from 20 mA cm⁻² to 150 mA cm⁻², see Figure 5e-5f and below). While stable electrochemical responses are recorded as the current density increases, the voltage efficiency decreases to the contrary, particularly for Step 1 in which the polarization loss from NiOR becomes significant. These results have been added in the revised manuscript (see highlights on Page 20 of revised of manuscript and Figure 5d-5e).

“We then showed the periodical swap between Step 1 and Step 2 at different current densities in Figure 5d. While relatively stable voltage responses were recorded from 20 mA cm⁻² to 150 mA cm⁻², the electrode voltage efficiencies shown in Figure 5e, for both CO and formate generation, decreased as the current density rose.”

Figure 5. (d) Periodical swap between Step 1 and Step 2 at 20, 50, 70, 100, 120, and 150 mA cm⁻² with the same DoD; (e) voltage efficiency of Step 1 and Step 2 at different current density.

2. It is noted that even under 250 mA cm⁻² (Fig. 4f), there is no O₂ product. But the demonstrated operation time of high current is quite short. At 50 mA cm⁻², the Step 1 (CO₂RR) can be operated smoothly for ~3h before OER starts (Fig. 4e). What about running Step 1 at a larger current density > 100 mA cm⁻²? How long can it stay stable without OER? There might be a trade-off between the polarization current density and the duty time. It would be great if these two factors of Step 1 can be further optimized to achieve an industrial-relevant level.

Response: This is a good question, we have now added additional experiment by running Step 1 at various current densities. We found the maximum possible one is 150 mA cm⁻² above which OER started to dominate and the voltage efficiency becomes unsatisfactory (<50%). In addition, we also studied the influence of the depth of charge (DoC, changing is defined as the Ni(OH)₂-to-NiOOH conversion) of Ni(OH)₂ on the oxidation reaction selectivity (NiOR vs OER). In the in-situ differential electrochemical mass spectrometry (DEMS) measurement where the O₂ production was monitored during NiOR (note that using online GC to study this is challenging as the amount of evolved gas is really small), we found that when the Ni(OH)₂ electrode is applied with a constant

current density equivalent to 150 mA cm^{-2} of the cell, (see Figure S26 and below), no OER occurred when DoC was below 20.8%. Yet, when the applied current density is equivalent to 40 mA cm^{-2} and below, no OER was observed even when DoC reached $>90\%$ (see Figure S27 and below). These results and corresponding discussion have been added in the revised manuscript (see highlights on Page 17 and Figure S26-S27).

“Further study indicated that the oxygen formation was also affected by DoC of the $\text{Ni}(\text{OH})_2$ electrode. When DoC was below 20.8%, no OER was observed at the current density equivalent to 150 mA cm^{-2} of the cell (see Figure S26). When the applied current density is equivalent to 40 mA cm^{-2} and below, no OER was monitored even when DoC reached $>90\%$ (see Figure S27).”

Figure S26. The ionic current response in the DEMS measurement in which $\text{Ni}(\text{OH})_2$ was used as the working electrode biased at a constant current density equivalent to 150 mA cm^{-2} of the cell.

Figure S27. The ionic current response in the DEMS measurement in which $\text{Ni}(\text{OH})_2$ was used as the working electrode biased at a constant current density equivalent to 40 mA cm^{-2} of the cell.

3. The authors target enhancing the total CO_2RR energy efficiency. The overall energy efficiency of this system needs to be provided and compared to all relevant alternative approaches. First on an energy basis, is the energy input here – including that required to electrocatalytically produce the H_2 less than the energy input required to produce the same output product via other means (including SOEC)? Then also cost and carbon comparisons for this case vs. the incumbent.

Response: This is a very good point and we have now carried out the techno-economic analysis to evaluate the materials cost of our H_2 -integrated CO_2RR (Route 1) based on the production of CO in comparison with two alternative approaches (Route 2: conventional CO_2RR flow cell combined with PEMFC; Route 3: solid-oxide electrolysis cells combined PEMFC). The schematic comparison

of three routes is shown in Figure S35. Our calculation indicates Route 1 costs ~\$23 and ~\$27 less than Route 2 and Route 3 per kilowatt, respectively. Meanwhile, the total energy efficiency (EE) of the above three routes were also evaluated based on the production of CO. The calculation indicates that Route 1 employed the highest EE value of 55.2% (Route 2: 35.8%; Route 3: 51.5%) at the same condition. The following discussion and calculation process were added in the Page 21 of revised manuscript and Page 60-64 of SI.

“We also carried out a preliminary techno-economic and energy efficiency analysis of CO₂RR coupling H₂ utilization using several representative technologies including flow cells and solid oxide electrolysis cells (SOECs, see all routes in Figure S35). Indeed, our cell is more costly than the conventional CO₂RR flow cells due to the use of mediator and Pt-containing GDE, yet it excludes the integration of PEMFC with expensive membrane and an order of magnitude higher Pt loading, yielding a much lower total cost. In terms of the overall energy conversion efficiency, our cell is comparable with the route involving highly-efficient SOEC while enjoying the capability of generating various C1-C3 products.”

Figure S35. (a) Schematic illustration of Route 1, Route 2 and Route 3 for combining CO₂RR with H₂-based energy storage; (b) materials cost comparison of three routes per kilowatt; (c) energy efficiency comparison of three routes

“Supplementary Note 1: Techno-economic analysis

We carried out the techno-economic analysis to evaluate the materials cost of our H₂-integrated CO₂RR (Route 1) based on the production of CO in comparison with two alternative approaches (Route 2: conventional CO₂RR flow cell combined with PEMFC; Route 3: solid-oxide electrolysis cells combined PEMFC). The schematic comparison of three routes is shown in Figure S35. Note that only the component of a single cell is considered for more accurate estimation. The

materials cost was calculated via the following equation:

$$C_{\text{materials}} = \frac{\sum_n p_n A_n}{P} \quad (7)$$

where P is the operation power of the cell, p_n is unit price of the cell component n , and A_n is the area of each cell component. Details are shown below.

The cost calculation for Route 1:

The total materials cost for Route 1 is $\sim \$78 \text{ kW}^{-1}$ which comprises of:

(1) Cost of Step 1. The cell components contain Zn-Cu foam GDE and Ni(OH)₂/NiOOH electrode (the unit prices were shown in Table S14). We selected the power at 50 mA cm^{-2} (2.15 V , 0.11 W cm^{-2}) as a reference. Thus, the calculated cost of conventional CO₂RR cell was $\sim \$64 \text{ kW}^{-1}$.

(2) Cost of Step 2. The cell component only contain Pt GDE and separator as other components were considered in Step 1, because this process is completed in the same cell with Step 1. The selected power was at 0.12 W cm^{-2} (maximum power of Step 2, according to the power density plots of Step 2 in Figure 5a). Thus, the calculated cost for Step 2 was $\sim \$14 \text{ kW}^{-1}$.

2. The cost calculation for Route 2:

The total materials cost for Route 2 is combined of that from the conventional CO₂RR cell and a PEMFC. The total cost for Route 2 is $\sim \$101 \text{ kW}^{-1}$ which comprises:

(1) PEMFC. The PEMFC components contain two Pt GDEs, and a piece of Nafion membrane (the unit prices were shown in Table S14). The selected power was at 0.66 W cm^{-2} (maximum power of PEMFC, according to the power density plots of H₂-air PEMFC in Figure S31). Thus, the calculated cost was $\sim \$82 \text{ kW}^{-1}$.

(2) Conventional CO₂ RR cell. The selected power was at 50 mA cm^{-2} (2.32 V , 0.12 W cm^{-2}). The cell components contain Zn-Cu foam GDE and Co₃O₄-Ni electrode (the unit prices were shown in Table S14). Thus, the calculated cost was $\sim \$19 \text{ kW}^{-1}$.

3. The cost calculation for Route 3:

The total materials cost for Route 3 is combined of that from a solid-oxide electrolysis cell (SOEC) and from a PEMFC. The total cost for Route 3 is $\sim \$105 \text{ kW}^{-1}$ which comprises of:

(1) PEMFC. The cost of this part is the same as that in Route 2.

(2) SOEC. As the SOEC generally operated in the current range of $0.5\text{-}1.5 \text{ A cm}^{-2}$, the selected power was at 500 mA cm^{-2} (1.2 V , 0.6 W cm^{-2})³⁴. The cell components refer to our previous work, which contains cathode electrode (made from NiO, ZrO₂, and corn), ZrO₂ electrolyte, and anode electrode (made from La_{0.8}Sr_{0.2}MnO_{3-x}) (the unit prices were shown in Table S14). Thus, the calculated cost was $\sim \$23 \text{ kW}^{-1}$.

Supplementary Note 2: Calculation of energy efficiency

The total energy efficiency (EE) of the above three routes were also evaluated based on the production of CO. The calculation indicates that Route 1 employed the highest EE value of 55.2% (Route 2: 35.8%; Route 3: 51.5%) at the same condition.

1. The EE calculation for Route 1:

The total EE was evaluated based on Step 1 for CO₂RR and Step 2 for “H₂-electricity” conversion.

(1) EE of Step 1. The EE of CO₂RR could be expressed as follows:

$$EE_1 = \frac{W_j}{W_{\text{input}}} \times 100\% \quad (8)$$

where W_{input} is electricity energy input, and W_j is the energy associated with the output product.

Then the EE could be written as follow:

$$EE_1 = \frac{n_j \times LHV_j}{E_c \times Q} \times 100\% \quad (9)$$

Where LHV_j is the lower heating value of product (CO: 282.98 kJ mol⁻¹; HCOOH: 209.82 kJ mol⁻¹); n_j is the mole number of products; E_c is the cell voltage; and Q is the consumed charge quantity. The selected current density was 50 mA cm⁻² (2.15 V). Thus, the calculated EE_1 of step 1 was 68.2%. (2) EE of Step 2. The EE of “H₂-electricity” conversion could be expressed as follows:

$$EE_2 = \frac{W_{output}}{W_{H_2}} \times 100\% \quad (10)$$

where W_{output} is electricity energy output, and W_{H_2} is the energy associated with the input H₂. Then the EE could be written as follow:

$$EE_2 = \frac{E_c \times Q}{n_{H_2} \times HHV_{H_2}} \times 100\% \quad (11)$$

where HHV_{H_2} is the higher heating value of H₂ (285.8 kJ mol⁻¹); n_{H_2} is the mole number of consumed H₂; and Q is the generated charge quantity. The selected current density was 50 mA cm⁻² (1.2 V). Thus, the calculated EE_2 of Step 2 was 81.0%.

Finally, the total EE for Route 1 was calculated to be 55.2% using the equation as follows:

$$EE_{total} = EE_1 \times EE_2 \quad (12)$$

2. The EE calculation for Route 2:

The total EE was evaluated based on both conventional CO₂RR cell and H₂ powered PEMFC, which is calculated to be 35.8%.

(1) EE of conventional CO₂RR cell. The EE of the cell for conventional CO₂RR cell in this work was calculated through equation (8-9) as well. The selected current density was also 50 mA cm⁻² (2.32 V). Thus, the calculated EE_1 of conventional CO₂RR cell was 63.2%.

(2) EE of PEMFC. The EE of the PEMFC in this work was calculated through equation (10-11) as well. We did not select the current density at the maximum power density but at 50 mA cm⁻² to with a higher voltage efficiency (0.84 V, see Figure S31). Thus, the calculated EE_2 of PEMFC was 56.7%.

3. The EE calculation for Route 3:

The total EE for Route 3 is combined of that from a SOEC and form a PEMFC, which is calculated as 51.5%.

(1) EE of SOEC. The EE of SOEC could also be calculated through equation (8). However, in SOEC solo mode, in addition to electricity income, there is the energy consumption for thermal load. Here, we assumed that the energy consumption for thermal load was from hydrogen combustion. Thus, equation (8) can be rewritten as:

$$EE_1 = \frac{n_j \times LHV_j}{E_c \times Q + n_{H_2} \times HHV_{H_2}} \times 100\% \quad (13)$$

The selected current density was 50 mA cm⁻² (0.15 V reported in literature³⁴). Thus, the calculated EE_2 of SOEC was 90.8%.

(2) EE of PEMFC. The EE of this part is the same as that in Route 2.”

Table S14. The cost of cell component.

Component	Cost	Units	Source
Zn-Cu foam GDE	12.8	\$ m ⁻²	a
Co ₃ O ₄ electrode	9.8	\$ m ⁻²	a
Ni(OH) ₂ /NiOOH	14.0	\$ m ⁻²	a
Pt GDE	16.0	\$ m ⁻²	a
Separator	2.0	\$ m ⁻²	PNAS 115, 11694-11699 (2018)
Nafion membrane	350	\$ m ⁻²	J. Power Sources 288, 187-198 (2015)
Cathodic electrode of SOEC	81.5	\$ m ⁻²	a
Anodic electrode of SOEC	53.8	\$ m ⁻²	a

a. based on the chemicals/materials cost in the Table S15.

Table S15. The cost of chemicals/materials.

Chemicals/materials	Cost	Units	Source
Carbon black	0.04	\$ kg ⁻¹	a
PTFE	10.3	\$ kg ⁻¹	b
Zn	8.0	\$ kg ⁻¹	c
Cu foam	2.4	\$ m ⁻²	c
Co	75.0	\$ kg ⁻¹	c
Ni foam	6.0	\$ m ⁻²	Nat. Energy 6, 517-528 (2021)
Ni	14.0	\$ kg ⁻¹	c
Pt	32035.0	\$ kg ⁻¹	c
NiO	14.0	\$ kg ⁻¹	c
ZrO ₂	45.8	\$ kg ⁻¹	d
Corn	0.19	\$ kg ⁻¹	e
La _{0.8} Sr _{0.2} MnO _{3-x}	7000.0	\$ kg ⁻¹	f
GDL	0.006	\$ m ⁻²	Renew. Sust. Energ. Rev. 154, 111807 (2022).

a. Taken from online report of “carbon black price trend and forecast” (<https://www.chemanalyst.com/>).

b. Taken from online report of “PTFE prices rising due to improvement in downstream market demand” (<https://www.chemanalyst.com/>).

c. Taken from the report of daily metal prices (<https://www.dailymetalprice.com/>).

d. Taken from online trade market. (https://www.made-in-china.com/products-search/hot-china-products/Zro2_Zirconia_Powder_Price.html)

e. Taken from online trade market. (<https://www.derthickscornmaze.com/interesting-about-corn/faq-what-is-the-price-of-corn-per-bushel-today.html>)

f. Taken from online trade market. (<https://www.sigmaaldrich.cn/CN/zh/product/aldrich/704261>)

Figure S31. Polarization curves and power density plots of H₂-O₂ and H₂-air PEMFC.

4. Some small errors need to be carefully checked. E.g. the figure caption order of Fig. S7.

Response: We have now carefully revised the manuscript and corrected the grammatic and typographic errors.

5. I'm not sure why there is a H₂ fuel cell in 1a, and the associated narrative. I suppose the concept is to show that OER and ORR are costly in both CO₂ electrolyzers and H₂ fuel cells... but these technologies are not analogous in the way that H₂ electrolyzers and fuel cells are. H₂ makes electricity in the fuel cell case, it would help make carbon-based products more efficiently in the proposed case, and these are very different outcomes.

Response: It is our fault that the main argument of this work is not well presented in the previous version, all statements that might mislead readers have been removed now. Our ideal is to include “hydrogen cycle” in the CO₂RR for efficiency enhancement as shown in the revised Figure 1a (also see below), the new process eliminates both OER and ORR processes in CO₂RR and in “hydrogen-electricity” conversion process, respectively. The following discussion have been added and highlighted on Page 4.

*“The renewable energy, to power CO₂RR, is intermittent in nature. This means the continuous operation of CO₂ electrolysis requires the integration of reliable and affordable energy storage solutions which are also critical for supporting the worldwide deployment of renewable energy sources. H₂ is often regarded as a clean and flexible energy carrier which can “store” surplus renewable energy via water electrolysis and “release” electricity using fuel cells. Indeed, powering CO₂ electrolysis using electricity generated in hydrogen fuel cells is a good solution in this context as shown in the flowchart of **Figure 1a**. One would note that oxygen reduction reaction (ORR) in fuel cells is the reverse process of OER in CO₂RR, potentially offering opportunities of “cancelling” both half reactions in the cascade process. This is particularly advantageous as both reactions are sluggish 4-electron/proton transfer processes, drastically decreasing the energy efficiency of the respective units.”*

Figure 1. (a) Illustrative comparison of CO₂RR electrolysis powered by renewable energies stored in the form of hydrogen via conventional and H₂-integrated routes.

6. Throughout the wording and grammar could be improved. I suggest, for instance, changing the title to “Integrating hydrogen conversion in CO₂ electrolysis without oxygen evolution”. “The sluggish oxygen reduction reaction was also eliminated” this is of course a big part of the low *V* achieved, so “also” is not right. “useless” oxygen catalysis. Etc. the writing requires professional proofreading.

Response: We very appreciate this valuable comment. The manuscript is now carefully revised, particularly in terms of wording and grammar check. Meanwhile, inspired by your suggestion, we have now updated the title as “Integrating hydrogen utilization in CO₂ electrolysis with minimized polarization loss”.

7. Figure quality must be improved to meet the standards and norms of the journal.

Response: We have now revised all figures in terms of content and quality (see the updated figures in the revised manuscript).

REVIEWER COMMENTS

Reviewer #1 (Remarks to the Author):

Authors significantly revised the manuscript to clarify the concept of integrated H₂ utilization in CO₂ electrolyzers by leveraging Ni(OH)₂ mediator. Detailed techno-economical analysis was appreciated. The revised manuscript does convey the merits of the engineering that was missing in the original manuscript.

Reviewer #2 (Remarks to the Author):

The authors have put a lot of effort in addressing the reviewer comments, and have done this adequately in my opinion. Therefore, I recommend publication of this revised manuscript

Reviewer #3 (Remarks to the Author):

I reviewed the responses to all reviewers and the revised MS. I appreciate the revision, however, I still do not agree with the fuel cell based comparison case on which this MS improves.

The comparison case (renewables->H₂->fuelcell->CO₂Rcell) is not established, and is not – in my view – a viable approach. The premise that we need energy storage to perform CO₂R does not make sense to me (it is more likely that CO₂R will need to use intermittent renewable electricity directly). So improving on that flawed base case scenario with a CO₂Rcell that takes H₂ directly is not compelling.

The two cases that ARE of interest are:

1 - The base case of renewables->CO₂Rcell with standard OER

2 – The author's case of step-a (renewables->H₂) and step-b (renewables and H₂->CO₂Rcell with authors system)

No fuel cells.

In addition, the long term operation is too short to provide confidence.

Comments and revisions of Reviewer 3:

1. I reviewed the responses to all reviewers and the revised MS. I appreciate the revision, however, I still do not agree with the fuel cell based comparison case on which this MS improves. The comparison case (renewables->H₂->fuelcell->CO₂Rcell) is not established, and is not – in my view – a viable approach. The premise that we need energy storage to perform CO₂R does not make sense to me (it is more likely that CO₂R will need to use intermittent renewable electricity directly). So improving on that flawed base case scenario with a CO₂Rcell that takes H₂ directly is not compelling.

The two cases that ARE of interest are:

- 1 - The base case of renewables->CO₂Rcell with standard OER
- 2 – The author’s case of step-a (renewables->H₂) and step-b (renewables and H₂->CO₂Rcell with authors system) No fuel cells.

Response: We appreciate this critical comment which is very reasonable. Now we have “disconnected” fuel cells with CO₂RR and revised the presentation (figures and discussions) in all relevant sections of the manuscript (see the highlights) and have the following routes:

1. Conventional CO₂RR + OER
2. H₂-integrated CO₂RR
3. Conventional “H₂-to-power” via fuel cells

Note that the fuel cell section is now only used as a benchmark for the overpotential comparison regarding hydrogen utilizations (HOR).

The following corrections on the background information are highlighted on **pp.4** and **Figure 1a**.
“It should be noted that on the “big picture” of renewable energy, in addition to chemical transformations such as CO₂RR, the integration of energy storage solution is another major constituent to address the intermittency and seasonal-variation problem. Hydrogen is often regarded as a clean and flexible energy carrier which can “store” surplus renewable energy and “release” electricity upon request. The annual hydrogen energy storage market is projected to increase seven-fold, compared with that in 2020, by 2030 to more than 35 GWh. Fuel cell is an ideal electrochemical device for “hydrogen-to-power” conversion, which involves the hydrogen oxidation reaction (HOR) and oxygen reduction reaction (ORR). Notably, ORR is the reverse process of OER in CO₂RR. Combining hydrogen conversion with CO₂RR potentially offers opportunities of “cancelling” both ORR- and OER-half reactions on the big picture (Figure 1a). This is energetically advantageous as both cancelled half reactions are sluggish 4-electron/proton transfer processes, drastically decreasing the energy efficiency of the respective process.”

Figure 1a. Illustrative comparison of H₂-integrated CO₂RR with typical CO₂RR and power-to-hydrogen-to-power via conventional routes on the big picture of renewable energy storage and

conversion.

The following corrections on the overpotential analysis are highlighted on **pp.22** and **Figure 5g**.

“Regarding the hydrogen utilization process, the conventional “H₂-to-power” using a PEMFC suffered 0.39 V polarization loss at 50 mA cm⁻² which was also significantly higher than that in Step 2 (0.15V). The detailed polarization losses of each electrode/process are also listed in Table S11. Thus, we concluded that H₂-integrated CO₂RR was able to substantially minimize overpotential losses in conventional CO₂RR and “H₂-to-power” processes.”

Figure 5g. Contributions of various polarization losses in H₂-integrated and conventional CO₂RR process at 50 mA cm⁻².

The following corrections on the techno-economic and energy efficiency analyses are highlighted on **pp.22-23** and relevant sections (supplementary note 2 and Figure S38) in the **supporting information**.

“In terms of the energy conversion efficiency, both CO₂RR and “H₂-to-power” in the new cell are advantageous as compared with conventional counterparts working at low temperatures. Although the energy efficiency cannot compete with SOEC operating at elevated temperatures (>600 °C), our cell enjoys the capability of generating various C1-C3 products.”

Figure S38. (a) Schematic illustration of Route 1, Route 2 and Route 3 for combining CO₂RR with “H₂-to-power”; (b) materials cost comparison of three routes per kilowatt; (c) energy efficiency of CO₂RR and “H₂-to-power” in various cells (H₂-intergrated CO₂RR cell, CO₂RR flow cell, SOEC, fuel cell) at 50 mA cm⁻².

2. In addition, the long term operation is too short to provide confidence.

Response: We agree with the reviewer on the stability test and have now carried out 100 h test for both CO₂-to-CO and CO₂-to-formate conversions. The spent catalysts together with the electrode were also analyzed using microscopic method. 5~6% voltage degradation in Step 1 and no voltage degradation in Step 2 were observed in both processes. The morphology of the catalysts and the GDE showed no change after the test.

The following revision are highlighted on **pp.21** and **Figures S32-S35** in the SI.

“We thus performed a 100 h longevity test for both CO₂-to-CO and CO₂-to-formate conversion as shown in Figures S32 and S33. The voltage degradation was trivial and only came from Step 1 in both processes (0.11 V and 0.12V for CO and formate generation, respectively). The spent Zn and Bi₂O₃ GEDs were analyzed using SEM. Original nanostructure of both catalysts were maintained after the test. In particularly, the gradient functional layer of the Zn based GDE remains intact, no cracks or delamination was observed on the SEM images shown in Figures S34 and S35.”

Figure S32. Longevity test of periodical swap between Step 1 and Step 2 at 50 mA cm^{-2} for CO generation.

Figure S32. Longevity test of periodical swap between Step 1 and Step 2 at 50 mA cm^{-2} for formate generation.

Figure S34. SEM images of Zn-Cu foam based GDE for (a) catalysts side and (b) gas diffusion layer side after longevity test.

Figure S35. SEM images of Bi_2O_3 -based GDE for (a) catalysts side and (b) gas diffusion layer side after longevity test.

Reviewers' comments:

Reviewer #3 (Remarks to the Author):

Re. point 2 – I appreciate the 100 hours of stability achieved, this meets the mark.

Re. point 1 – The inclusion of fuel cells still muddies the case here. A few clarifying points:

- Apples-to-apples comparison is critical here. Right now, with fuel cells included, you have a case with CO₂R *chemical products* compared (on materials cost, efficiency, etc basis) with a fuel cell case that produces *electricity*. These cannot be compared side-by-side because the end points are not the same.

- The new Figure 1a highlights wasted half-reactions in the comparison cases – but there is *also* a wasted half reaction in the production of H₂ from renewable energy. The renewable energy -> H₂ step should have this wasted half reaction shown, and the H for the H-integrated CO₂RR line should come from this not directly from renewable energy. Thus it would be clear that there are wasted half reactions in both cases of interest (as communicated previously)

o 1 - The base case of renewables->CO₂Rcell with standard OER

o 2 – The author's case of step-a (renewables->H₂) and step-b (renewables and H₂->CO₂Rcell with authors system) No fuel cells.

- The key metric on which to compare 1 and 2 is GJ/tonne-product total (for the same product produced). The MS is not clear on this. It is not clear if the voltages are full cell voltages (i.e. real indicators of energy input). The key question is, for a given amount of electricity input, which route 1 or 2 would maximize output product.

Comments and revisions of Reviewer 3:

1. Re. point 2 – I appreciate the 100 hours of stability achieved, this meets the mark.

Response: We again thank the referee for this comment.

2. Re. point 1 – The inclusion of fuel cells still muddies the case here. A few clarifying points:

- Apples-to-apples comparison is critical here. Right now, with fuel cells included, you have a case with CO₂R *chemical products* compared (on materials cost, efficiency, etc basis) with a fuel cell case that produces *electricity*. These cannot be compared side-by-side because the end points are not the same.

- The new Figure 1a highlights wasted half-reactions in the comparison cases – but there is *also* a wasted half reaction in the production of H₂ from renewable energy. The renewable energy → H₂ step should have this wasted half reaction shown, and the H for the H-integrated CO₂RR line should come from this not directly from renewable energy. Thus, it would be clear that there are wasted half reactions in both cases of interest (as communicated previously)

o 1 - The base case of renewables-→CO₂Rcell with standard OER

o 2 – The author’s case of step-a (renewables-→H₂) and step-b (renewables and H₂-→CO₂Rcell with authors system) No fuel cells.

- The key metric on which to compare 1 and 2 is GJ/tonne-product total (for the same product produced). The MS is not clear on this. It is not clear if the voltages are full cell voltages (i.e. real indicators of energy input). The key question is, for a given amount of electricity input, which route 1 or 2 would maximize output product.

Response: We sincerely appreciate this clear and rational comment which really shows the importance of peer review process. We admit that the so-called case 1 and case 2 above, together with the energy consumption analysis, is scientifically better than our previous model. In the revised manuscript, we have completely followed them (see **Figure 1a** and below).

To accurately evaluate the energy input during water electrolysis, we built an alkaline water electrolyzer (AWE) and a solid oxide electrolysis cell (SOEC). Our key finding is that the H₂-integrated CO₂RR + water electrolysis, despite of the increased system complexity, cuts the total energy consumption in all cases (neutral or alkaline; AWE or SOEC, various current densities etc.) compared with the conventional counterparts. Particularly in neutral case the decrease can reach 42%. This is because **(1) OER** in conventional CO₂RR reactor, with an overpotential often higher than 0.52 V (50 mA cm⁻²)^{ref37}, is “**transferred**” to a water electrolyzer with **thermodynamically/kinetically more favored reaction conditions**, and **(2) the elimination of anodic CO₂ recovery**. Note that we used “potential” to indicate the potential of an electrode, and used “voltage” to indicate that of a full cell. To make it clearer, we have changed “voltage” to “cell voltage” in the manuscript. (see revised **Figure 5h and i** and below)

Please also note that that our manuscript has been thoroughly revised in both sections of both “Introduction” and “Results & discussions” with highlights in yellow. The SI now also contains various revisions (see **Tables S11 and S14-S19**; **Figures S32-S34 and S41**; **Supplementary Notes 2 and 3**). They are also attached below for your convenience.

Figure S32. (a) SEM cross-sectional image of the electrode-supported SOEC; (b) polarization plots of water electrolysis at 800 °C and 850 °C, the inlet feedstock is 50 vol% H₂O balanced by H₂.

Figure S33. The design and photograph of a home-built solid oxide cells (SOC) test system for both SOEC and solid oxide fuel cells (SOFC) test.

Figure S34. (a) Schematic structure of a home-built alkaline water electrolyzer (AWE); (b) the polarization plot of water electrolysis in 6M KOH at 85 °C and ambient pressure.

Figure S41. (a) Schematic illustration of H₂-integrated CO₂RR coupled with water electrolysis and conventional CO₂RR; (b) preliminary techno-economic analysis of H₂-integrated CO₂RR coupled with water electrolysis and conventional CO₂RR based on CO production, only materials cost was considered; comparison of energy efficiency between H₂-integrated CO₂RR coupled with water electrolysis and conventional CO₂RR at 50 mA cm⁻² in (c) alkaline and (d) neutral electrolyte.

Table S14. The comparison of energy consumption between H₂-integrated CO₂RR coupled with water electrolysis and conventional CO₂RR for CO production in 1 M KOH at different current densities.

	Current density (mA cm ⁻²)	20	50	70	100
H₂-integrated CO₂RR+SOEC	Cell voltage (V) - H ₂ -integrated CO ₂ RR	0.51	0.87	1.05	1.26
	CO FE (%)	70.6	71.3	73.1	75.6
	Electricity-CO ₂ RR (GJ per tonne CO)	5.0	8.4	9.9	11.5
	Cell voltage (V) -SOEC	0.930	0.940	0.945	0.951
	Electricity-water electrolysis (GJ per tonne CO)	9.1	9.1	8.9	8.7
	Energy consumption (GJ per tonne CO)	14.1	17.5	18.8	20.2
H₂-integrated CO₂RR+AWE	Cell voltage (V) - H ₂ -integrated CO ₂ RR	0.51	0.87	1.05	1.26
	CO FE (%)	70.6	71.3	73.1	75.6
	Electricity (GJ per tonne CO)	5.0	8.4	9.9	11.5
	Cell voltage (V) -AWE	1.39	1.43	1.46	1.48
	Electricity-water electrolysis (GJ per tonne CO)	13.6	13.8	13.8	13.5
	Energy consumption (GJ per tonne CO)	18.6	22.2	23.7	25.0
conventional CO₂RR	Cell voltage (V) -conventional CO ₂ RR	2.01	2.34	2.50	2.64
	CO FE (%)	65.5	71.3	72.5	74.6
	Energy consumption (GJ per tonne CO)	21.1	22.4	23.8	24.7

Table S15. The comparison of energy consumption between H₂-integrated CO₂RR coupled with water electrolysis and conventional CO₂RR for formate production in 1 M KOH at different current densities.

	Current density (mA cm ⁻²)	20	50	70	100
H₂-integrated CO₂RR+SOEC	Cell voltage (V) - H ₂ -integrated CO ₂ RR	0.59	0.89	1.05	1.25
	Formate FE (%)	87.8	93.8	94.8	94.3
	Electricity-CO ₂ RR (GJ per tonne formate)	3.2	4.6	5.3	6.4
	Cell voltage (V) -SOEC	0.930	0.940	0.945	0.951
	Electricity-water electrolysis (GJ per tonne formate)	5.1	4.8	4.8	4.9
	Energy consumption (per tonne formate)	8.3	9.4	10.1	11.3
H₂-integrated CO₂RR+AWE	Cell voltage (V) - H ₂ -integrated CO ₂ RR	0.59	0.89	1.05	1.25
	Formate FE (%)	87.8	93.8	94.8	94.3
	Electricity (GJ per tonne formate)	3.2	4.6	5.3	6.4
	Cell voltage (V) -AWE	1.39	1.43	1.46	1.48
	Electricity-water electrolysis (GJ per tonne formate)	7.6	7.4	7.4	7.6
	Energy consumption (GJ per tonne formate)	10.8	12.0	12.7	14.0
conventional CO₂RR	Cell voltage (V) -conventional CO ₂ RR	2.10	2.34	2.50	2.64

	Formate FE (%)	87.8	92.7	93.3	93.6
	Energy consumption (GJ per tonne formate)	11.6	12.2	12.9	13.6

Table S16. The comparison of energy consumption between H₂-integrated CO₂RR coupled with water electrolysis and conventional CO₂RR in 1 M KOH when water electrolyzer was operated at the industrially-relevant conditions.

	CO production		Formate production	
H₂-integrated CO₂RR+SOEC	Cell voltage (V) - H ₂ -integrated CO ₂ RR	0.87	Cell voltage (V) - H ₂ -integrated CO ₂ RR	0.89
	CO FE (%)	71.3	Formate FE (%)	93.8
	Electricity-CO ₂ RR (GJ per tonne CO)	8.4	Electricity-CO ₂ RR (GJ per tonne formate)	4.6
	Cell voltage (V) -SOEC	1.28	Cell voltage (V) -SOEC	1.28
	Electricity-water electrolysis (GJ per tonne CO)	12.4	Electricity-water electrolysis (GJ per tonne formate)	6.6
	Energy consumption (GJ per tonne CO)	20.8	Energy consumption (GJ per tonne formate)	11.2
H₂-integrated CO₂RR+AWE	Cell voltage (V) - H ₂ -integrated CO ₂ RR	0.87	Cell voltage (V) - H ₂ -integrated CO ₂ RR	0.89
	CO FE (%)	71.3	Formate FE (%)	93.8
	Electricity (GJ per tonne CO)	8.4	Electricity (GJ per tonne formate)	4.6
	Cell voltage (V) -AWE	1.47	Cell voltage (V) -AWE	1.47
	Electricity-water electrolysis (GJ per tonne CO)	14.2	Electricity-water electrolysis (GJ per tonne formate)	7.6
	Energy consumption (GJ per tonne CO)	22.6	Energy consumption (GJ per tonne formate)	12.2
conventional CO₂RR	Cell voltage (V) -conventional CO ₂ RR	2.34	Cell voltage (V) -conventional CO ₂ RR	2.34
	CO FE (%)	71.3	Formate FE (%)	92.7
	Energy consumption (GJ per tonne CO)	22.4	Energy consumption (GJ per tonne formate)	12.2

Table S17. The comparison of energy consumption between H₂-integrated CO₂RR coupled with water electrolysis and conventional CO₂RR for CO production in 1 M KHCO₃ at different current densities.

	Current density (mA cm ⁻²)	20	50	70	100
H₂-integrated CO₂RR+SOEC	Cell voltage (V) - H ₂ -integrated CO ₂ RR	0.93	1.21	1.38	1.59
	CO FE (%)	60.6	63.3	66.1	70.6
	Electricity-CO ₂ RR (GJ per tonne CO)	10.6	13.2	14.4	15.5
	Anode gas separation (GJ per tonne CO)	0	0	0	0
	Cell voltage (V) -SOEC	0.930	0.940	0.945	0.951
	Electricity-water electrolysis (GJ per tonne CO)	10.6	10.2	9.9	9.3
	Energy consumption (GJ per tonne CO)	21.2	23.4	24.3	24.8
H₂-integrated CO₂RR+AWE	Cell voltage (V) - H ₂ -integrated CO ₂ RR	0.93	1.21	1.38	1.59
	CO FE (%)	60.6	63.3	66.1	70.6

	Electricity (GJ per tonne CO)	10.6	13.2	14.4	15.5
	Anode gas separation (GJ per tonne CO)	0	0	0	0
	Cell voltage (V) -AWE	1.39	1.43	1.46	1.48
	Electricity-water electrolysis (GJ per tonne CO)	15.8	15.6	15.2	14.4
	Energy consumption (GJ per tonne CO)	26.4	28.8	29.6	29.9
conventional CO₂RR	Cell voltage (V) -conventional CO ₂ RR	2.46	2.66	2.84	3.01
	CO FE (%)	55.8	62.8	65.5	70.6
	Electricity (GJ per tonne CO)	30.4	29.2	29.9	29.4
	Anode gas separation (GJ per tonne CO)	11.3	10.0	9.6	8.9
	Energy consumption (GJ per tonne CO)	41.7	39.2	39.5	38.3

Table S18. The comparison of energy consumption between H₂-integrated CO₂RR coupled with water electrolysis and conventional CO₂RR for formate production in 1 M KHCO₃ at different current densities.

	Current density (mA cm ⁻²)	20	50	70	100
H₂-integrated CO₂RR+SOEC	Cell voltage (V) - H ₂ -integrated CO ₂ RR	0.84	1.2	1.44	1.79
	Formate FE (%)	81.8	85.8	89.8	89.3
	Electricity-CO ₂ RR (GJ per tonne formate)	5.0	6.7	7.7	9.7
	Anode gas separation (GJ per tonne formate)	0	0	0	0
	Cell voltage (V) -SOEC	0.930	0.940	0.945	0.951
	Electricity-water electrolysis (GJ per tonne formate)	5.5	5.3	5.1	5.1
	Energy consumption (GJ per tonne formate)	10.5	12.0	12.8	14.8
H₂-integrated CO₂RR+AWE	Cell voltage (V) - H ₂ -integrated CO ₂ RR	0.84	1.2	1.44	1.79
	Formate FE (%)	81.8	85.8	89.8	89.3
	Electricity (GJ per tonne formate)	5.0	6.7	7.7	9.7
	Anode gas separation (GJ per tonne formate)	0	0	0	0
	Cell voltage (V) -AWE	1.39	1.43	1.46	1.48
	Electricity-water electrolysis (GJ per tonne formate)	8.2	8.0	7.8	8.0
	Energy consumption (GJ per tonne formate)	13.2	14.7	15.5	17.7
conventional CO₂RR	Cell voltage (V) -conventional CO ₂ RR	2.36	2.7	2.9	3.2
	Formate FE (%)	81.8	84.7	88.8	89.3
	Electricity (GJ per tonne formate)	13.9	15.4	15.8	17.3
	Anode gas separation (GJ per tonne formate O)	5.4	5.2	5.0	4.9
	Energy consumption (GJ per tonne formate)	19.3	20.6	20.8	22.2

Table S19. The comparison of energy consumption between H₂-integrated CO₂RR coupled with water electrolysis and conventional CO₂RR in 1 M KHCO₃ when water electrolyzer was operated at the industrially-relevant conditions.

	CO production		Formate production	
H₂-integrated CO₂RR+SOEC	Cell voltage (V) - H ₂ -integrated CO ₂ RR	1.21	Cell voltage (V) - H ₂ -integrated CO ₂ RR	1.2
	CO FE (%)	63.3	Formate FE (%)	85.8
	Electricity-CO ₂ RR (GJ per tonne CO)	13.2	Electricity-CO ₂ RR (GJ per tonne formate)	6.7
	Anode gas separation (GJ per tonne CO)	0	Anode gas separation (GJ per tonne formate)	0
	Cell voltage (V) -SOEC	1.28	Cell voltage (V) -SOEC	1.28
	Electricity-water electrolysis (GJ per tonne CO)	13.9	Electricity-water electrolysis (GJ per tonne formate)	7.2
	Energy consumption (GJ per tonne CO)	27.1	Energy consumption (GJ per tonne formate)	13.9
H₂-integrated CO₂RR+AWE	Cell voltage (V) - H ₂ -integrated CO ₂ RR	1.21	Cell voltage (V) - H ₂ -integrated CO ₂ RR	1.2
	CO FE (%)	63.3	Formate FE (%)	85.8
	Electricity (GJ per tonne CO)	13.2	Electricity (GJ per tonne formate)	6.7
	Anode gas separation (GJ per tonne CO)	0	Anode gas separation (GJ per tonne formate)	0
	Cell voltage (V) -AWE	1.47	Cell voltage (V) -AWE	1.47
	Electricity-water electrolysis (GJ per tonne CO)	16.0	Electricity-water electrolysis (GJ per tonne formate)	8.3
	Energy consumption (GJ per tonne CO)	29.2	Energy consumption (GJ per tonne formate)	15.0
conventional CO₂RR	Cell voltage (V) -conventional CO ₂ RR	2.66	Cell voltage (V) -conventional CO ₂ RR	2.70
	CO FE (%)	62.8	Formate FE (%)	84.7
	Electricity (GJ per tonne CO)	29.2	Electricity (GJ per tonne formate)	15.4
	Anode gas separation (GJ per tonne CO)	10.0	Anode gas separation (GJ per tonne formate)	5.2
	Energy consumption (GJ per tonne CO)	39.2	Energy consumption (GJ per tonne formate)	20.6

REVIEWERS' COMMENTS

Reviewer #3 (Remarks to the Author):

I appreciate the revision, and am convinced of the value of the approach and that it warrants publication. You might consider - in the text only - an additional comparison case of SOEC for CO₂ to CO. We usually assume this to be 20GJ/t_CO. Perhaps you could comment in the text how your approach improves on this single step SOEC. I suppose with the AWE route you are all low temperature and perhaps more stable and scalable.

Comments and revisions of Reviewer 3:

I appreciate the revision, and am convinced of the value of the approach and that it warrants publication. You might consider - in the text only - an additional comparison case of SOEC for CO₂ to CO. We usually assume this to be 20GJ/t_CO. Perhaps you could comment in the text how your approach improves on this single step SOEC. I suppose with the AWE route you are all low temperature and perhaps more stable and scalable.

Response: We again thank the referee your comment. The suggested discussion has been added and highlighted on Page 18 (also see below).

“Indeed, direct CO₂-to-CO conversion in SOEC consumes even less energy (e.g., ~13.5 GJ⁵²), but the low-temperature CO₂ electrolysis enjoys the capability of generating various C1-C3 products.”